# Integrative multi-omics reveals a regulatory and exhausted T-cell landscape in CLL and identifies galectin-9 as an immunotherapy target

L. Llaó-Cid [1,16], JKL Wong[1,16], I. Fernandez Botana[2], Y. Paul [1], M. Wierz[2], L-M Pilger[1,3], A. Floerchinger[1,3], CL Tan [4], S. Gonder[2], G. Pagano[2], M. Chazotte [5], K. Bestak [5], C. Schifflers [1], M. Iskar [1], T. Roider [6], F. Czernilofsky [6,7,8], P-M Bruch [6], JP Mallm [9], A. Cosma [10], DE Campton[11], E. Gerhard-Hartmann[12], A. Rosenwald[12], D. Colomer [13], E. Campo [13], D. Schapiro [5,14,15], EW Green [4], S. Dietrich [6], P. Lichter[1], E. Moussay [2,16], J. Paggetti [2,16], M. Zapatka [1,16] & M. Seiffert [1,16] ✉

T-cell exhaustion contributes to immunotherapy failure in chronic lymphocytic leukemia (CLL). Here, we analyze T cells from CLL patients' blood, bone marrow, and lymph nodes, as well as from a CLL mouse model, using single-cell RNA sequencing, mass cytometry, and tissue imaging. T cells in CLL lymph nodes show the most distinct profiles, with accumulation of regulatory T cells and CD8+ T cells in various exhaustion states, including precursor ($T_{PEX}$) and terminally exhausted ($T_{EX}$) cells. Integration of T-cell receptor sequencing data and use of the predicTCR classifier suggest an enrichment of CLL-reactive T cells in lymph nodes. Interactome studies reveal potential immunotherapy targets, notably galectin-9, a TIM3 ligand. Inhibiting galectin-9 in mice reduces disease progression and TIM3+ T cells. Galectin-9 expression also correlates with worse survival in CLL and other cancers, suggesting its role in immune evasion and potential as a therapeutic target.

Failure of response to immune checkpoint inhibitors (ICI) is commonly seen in cancer patients with unclear causes and predictors, limiting the use of these immunotherapy approaches. B-cell lymphomas develop in lymph nodes (LNs), the site of T-cell priming and activation in infections and cancer. Even though there is a constant interaction of malignant B cells with CD4+ and CD8+ T cells in this tissue, adaptive immune control fails and response to ICI is very limited in patients with B-cell lymphomas. Very low response rates to anti-PD1 antibodies were observed in patients with chronic lymphocytic leukemia (CLL)[1,2], a malignancy of mature B cells that accumulate in blood, LNs, and bone marrow. In contrast to patients with acute lymphocytic leukemia (ALL), also response rates to CD19 CAR-T-cell therapy remain below the

expectations in CLL[3]. A lack of fit and functional T cells is discussed as a main limitation explaining the failure of immunotherapy responses in CLL and beyond[4,5]. Chronic exposure of T cells to tumor (neo)antigens and an immune suppressive microenvironment lead to their exhaustion and dysfunction[6]. In addition, current treatment regimens for CLL were shown to impact T-cell fitness[7]. Whereas terminally exhausted T cells, characterized by high PD1 expression levels and expression of TOX, fail to reactivation by ICI, their precursor state that expresses lower levels of PD1 and is positive for TCF-7, harbors self-renewal capacity and the ability to control tumor development upon ICI treatment[8–10]. The presence of these cell states has been confirmed in CLL patients and mouse models[11], even though a characterization of

T cells in CLL LNs is lacking due to the limited access to such samples. Current efforts aim at characterizing mechanisms of T-cell exhaustion in cancer and at identifying therapeutic targets to overcome this limitation.

Here, we used single-cell RNA sequencing (scRNA-seq) and mass cytometry (CyTOF) of LNs, peripheral blood (PB) and bone marrow (BM) samples of patients with CLL, as well as reactive lymph nodes (rLN) as control, combined with multiplex imaging of LN tissue sections and microarrays to characterize the distribution, phenotype, and function of T cells under chronic exposure to cancer. We show that CLL LNs constitute a unique niche where clonally expanded CD8$^+$ T cells expressing CD39 and with an exhausted phenotype are enriched in comparison to reactive LNs, and are predicted to be CLL-reactive. We further defined cells with a precursor exhausted signature that accumulate in CLL LNs, as well as several types of regulatory T cells. The results of interactome analyses led us to target galectin-9 in a CLL mouse model, which resulted in improved T-cell function and an attenuated tumor development. Our analyses link galectin-9 expression with the survival of patients with CLL, kidney or brain tumors.

## Results

### Definition of the T-cell landscape in CLL using mass cytometry

To comprehensively characterize the T-cell compartment associated with CLL, a large-scale and high-dimensional analysis of T cells from CLL patients as well as non-cancer controls was conducted using mass cytometry, scRNA-seq, and multiplex imaging of tissue sections followed by integrative data analysis (Fig. 1A). The patient cohort reflects the heterogeneity of CLL in respect to cell-of-origin, including both *IGHV* mutated ($n = 10$) and unmutated ($n = 10$) cases, as well as patients with the prevalent genetic aberrations (Supplementary Data 1). We performed mass cytometry profiling of T cells from 22 CLL LNs, plus 7 paired PB and 3 paired BM samples, as well as 13 reactive LNs (rLNs) using a panel of 42 antibodies (Supplementary Data 2) designed to identify and characterize naïve, memory, effector, regulatory, and exhausted T cells. The analysis comprised $5.29 \times 10^6$ T cells, with a median of 51,937 cells per sample (Supplementary Fig. 1A–C). Unsupervised graph-based clustering based on the differential expression of the analyzed proteins grouped cells into 30 clusters which are presented as uniform manifold approximation and projection (UMAP) plot (Fig. 1B, C, Supplementary Fig. 2A). This approach led to the identification of 15 CD4$^+$ T-cell clusters, 9 CD8$^+$ T-cell clusters, 4 double-negative T-cell clusters, and 2 clusters with a mixture of CD4$^+$ and CD8$^+$ T cells (Fig. 1D, Supplementary Fig. 2B, C, and Supplementary Data 3). The expression of CD45RA, CD45RO, and CCR7 allowed for a general classification of T cells into naïve (T$_N$, CD45RA$^+$ CD45RO$^-$ CCR7$^+$), central memory (T$_{CM}$, CD45RA$^-$ CD45RO$^+$ CCR7$^+$), effector memory (T$_{EM}$, CD45RA$^-$ CD45RO$^+$ CCR7$^-$) and effector memory cells re-expressing CD45RA (T$_{EMRA}$, CD45RA$^+$ CD45RO$^-$ CCR7$^-$). The CD8$^+$ T-cell subsets comprised 1 naïve (CD8 T$_N$), 1 central memory (CD8 T$_{CM}$), 3 subsets with a short-lived effector cell (SLEC) phenotype expressing TBET and KLRG1 (CD8 T$_{EM}$ TBET, CD8 T$_{EMRA}$ TBET, and CD8 T$_{EMRA}$ CD56)[12,13], and 4 effector memory subsets with a higher expression of the cytotoxic molecule granzyme K (GZMK) and an increasing expression of exhaustion-related molecules such as the inhibitory receptors PD1 and TIGIT, the transcription factors EOMES and TOX, and the ectoenzymes CD38 and CD39 (CD8 T$_{EM}$ GZMK, CD8 T$_{EM}$ TBET GZMK, CD8 T$_{EX}$ CD38, and CD8 T$_{EX}$ CD39)[14]. The CD4$^+$ T-cell subsets comprised 1 naïve (CD4 T$_N$), 3 central memory (CD4 T$_{CM1}$ CCR7, CD4 T$_{CM2}$ CD25, and CD4 T$_{CM3}$), 1 follicular helper (T$_{FH}$), and 6 effector memory subsets. Of these, two expressed TBET and KLRG1, resembling SLEC subsets (CD4 T$_{EM}$ CD56, and CD4 T$_{EM}$ TBET), while 3 of them expressed higher levels of PD1 and TIGIT (CD4 T$_{EM}$ PD1), CTLA4 and CD38 (CD4 T$_{EM}$ CTLA4), or CD39 (CD4 T$_{EM}$ CD39). The last CD4 T$_{EM}$ subset expressed EOMES, GZMK and PD1 (CD4 T$_{EM}$ GZMK), and shares therefore characteristics of T regulatory type 1 cells (T$_R$1) and CD4$^+$

T cells with cytotoxic properties[15,16]. In addition, FOXP3 expression identified 4 T$_{REG}$ clusters within the CD4$^+$ T cells, 2 with a central memory phenotype (CD4 T$_{REG-CM1}$ and CD4 T$_{REG-CM2}$), and 2 subsets with an activated inhibitory phenotype (CD4 T$_{REG}$ PD1 and CD4 T$_{REG}$ CD39). CD4 and CD8 double-negative (DN) CD3$^+$ T cells comprised 2 clusters with a phenotype similar to CD8 SLEC subsets (DN T$_{EM}$ KLRG1 and DN T$_{EMRA}$ TBET), and 2 cytotoxic clusters expressing GZMK and EOMES as well as CD38 (DN T$_{EMRA}$ CD38) or the transcription factor HELIOS (DN T$_{EM}$ HELIOS). A KI67$^+$ proliferative subset (T$_{PR}$), as well as a cluster with high expression of ICOS (T$_{EM}$ ICOS) were both composed of CD4$^+$ and CD8$^+$ T cells.

To identify relationships between the T-cell populations present in the CLL LNs, we correlated the subset frequencies of all individual CLL LN samples ($n = 20$, excluding 2 duplicate samples, Fig. 1E). Naïve CD4$^+$ and CD8$^+$ T cells as well as CD4$^+$ and CD8$^+$ T cells with a SLEC phenotype were significantly associated with each other, indicating that the LN tumor microenvironment (TME) is similarly influencing the differentiation of these cell states both in CD4$^+$ and CD8$^+$ T cells.

Spatial analyses of immune cell populations by multiplex immunofluorescence staining of a tissue microarray (TMA) of 42 CLL LN samples confirmed the presence and colocalization of malignant B cells with T cells in the tissue (Fig. 2A, Supplementary Fig. 3A, B). Quantification of the frequencies of the major immune cell subsets in the tissues revealed a positive correlation for CD8$^+$ PD1$^-$ and CD8 PD1$^+$ T cells, for myeloid cells and CD4 T$_{REG}$, and for CD4 T$_{REG}$ and CD8$^+$ T cells (Fig. 2B, Supplementary Fig. 3C) which is in line with a role of CD4 T$_{REG}$ in promoting the accumulation of dysfunctional CD8$^+$ T cells[17]. We next quantitatively assessed the spatial cell-cell interactions by calculating the enrichment of pairwise interacting cell types for all cores using Giotto[18]. This confirmed an enrichment of physical interactions between CD4 T$_{REG}$ and CD8 PD1$^+$ T cells as well as CD4 PD1$^+$ or PD1$^-$ T cells (Fig. 2C, D, Supplementary Fig. 3D) suggesting that T$_{REG}$ limit the activity of CD4 and CD8 T cells in the CLL LNs.

### CLL LNs are enriched with regulatory T cells and exhausted cytotoxic T cells

We next examined the distribution of T-cell immunophenotypes across the three different tissues analyzed in CLL. The sample composition of PB and BM was similar, but differed from LN samples (Fig. 3A, B, Supplementary Fig. 4A), suggesting that tissue cues significantly determine T-cell composition. PB and BM samples were enriched in SLEC T cells, such as CD8 T$_{EM}$ TBET, CD4 T$_{EM}$ TBET, and DN T$_{EMRA}$ TBET, while CLL LNs contained higher frequencies of exhausted T cells (CD8 T$_{EX}$ CD39 and CD8 T$_{EX}$ CD38), CD4 T$_{EM}$ GZMK cells, CD4 T$_{FH}$ cells, several CD4 T$_{REG}$ subsets (CD4 T$_{REG-CM1}$, CD4 T$_{REG}$ PD1, and CD4 T$_{REG}$ CD39), as well as proliferating cells (T$_{PR}$) (Fig. 3C). Even though the abundances among cell clusters within the tissue types correlated, no significant associations of the frequencies of cell subsets between PB and LN samples of the same patient were identified, underscoring a differential composition of T-cell subsets between tissues (Supplementary Fig. 4B).

The comparison of cell subset frequencies between CLL LNs and rLNs revealed that higher percentages of cytotoxic CD8 T$_{EM}$ GZMK as well as exhausted CD8$^+$ T cells (CD8 T$_{EX}$ CD39 and CD8 T$_{EX}$ CD38) were present in CLL LNs (Fig. 3D). Within the CD4 compartment, CD4 T$_{EM}$ GZMK cells and all CD4 T$_{REG}$ clusters (CD4 T$_{REG-CM1}$, CD4 T$_{REG-CM2}$, CD4 T$_{REG}$ PD1, and CD4 T$_{REG}$ CD39) were also enriched in CLL LNs, in addition to proliferating T$_{PR}$ cells and DN T$_{EM}$ HELIOS cells (Fig. 3D). The main enriched cell types in rLNs versus CLL LNs were naïve CD4 and CD8 T cells as well as CD4 T$_{FH}$ cells (Supplementary Fig. 4C). A comparative analysis of activation and exhaustion markers on CD8 T$_{EM}$ cells revealed significantly higher CD39 expression in CLL LNs in comparison to rLNs, and a tendency towards higher expression levels of CTLA4, OX40, and TOX in CLL LNs (Fig. 3E). Collectively, these results suggest that CLL LNs constitute a distinct niche where malignant B-cell

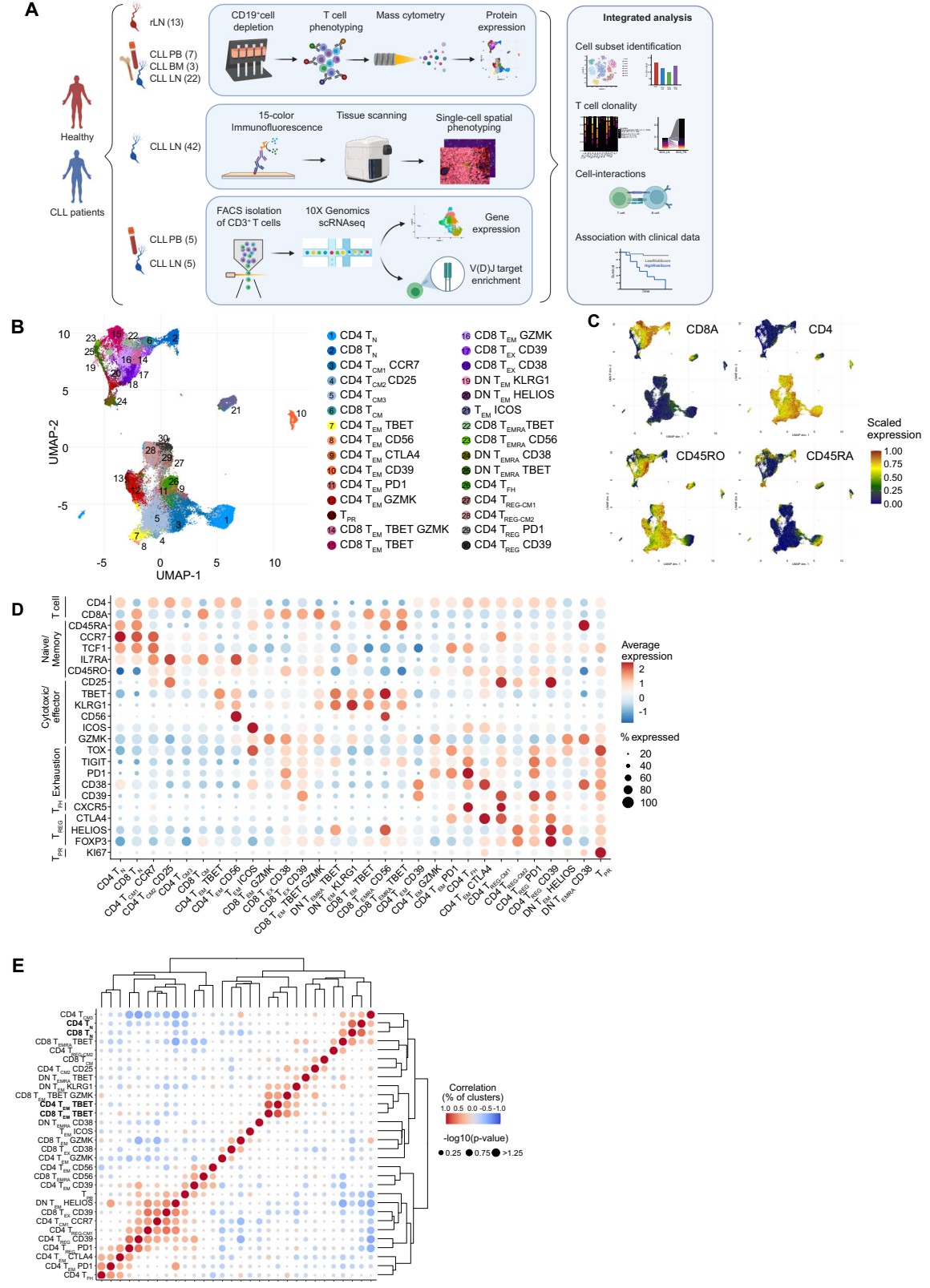

accumulation induces an immunosuppressive microenvironment with an enrichment of regulatory T cells and exhausted cytotoxic T cells.

Next, we examined the relation of clinical features to a distinct T-cell composition in the CLL LNs. We detected a strong negative association between the age of the patients and the frequency of naïve CD8+ T cells (Supplementary Fig. 4D) which is in line with published data[19]. In addition, the frequencies of 2 CD4 T$_{REG}$ clusters (CD4 T$_{REG}$-CM2 and CD4 T$_{REG}$ PD1) showed a positive correlation with age (Supplementary Fig. 4E, F). We did not observe significant associations between the frequencies of T-cell clusters with sex, clinical stage, or outcome of the patients according to the adjusted p-values after multiple comparisons (Supplementary Data 4). However, significant associations between several cell clusters and tumor load or clinical stage were detected before multiple testing (Supplementary Data 4; p-

**Fig. 1 | Profiling of the T-cell landscape in CLL tissues at the single-cell resolution. A** Graphical overview of the study design. Mass cytometry analyses were performed on 13 reactive lymph node (rLN), 7 CLL peripheral blood (PB), 3 CLL bone marrow (BM), and 22 CLL LN samples, 15-color multiplex immunostaining was applied in an additional data set of 42 CLL LNs, and 5 paired CLL LN and PB samples were analyzed using scRNA-seq and TCR-seq. Created in BioRender. Zapatka, M. (2025) https://BioRender.com/deezga2. **B** UMAP plot of 5.2 ×10[6] CD3[+] T cells from 13 rLN, 7 CLL PB, 3 CLL BM and 22 CLL LN samples analyzed by mass cytometry identifying 30 clusters, including 15 for CD4[+] T cells, 9 for CD8[+] T cells, 2

clusters containing both CD4[+] and CD8[+] T cells, and 4 CD4 and CD8-double-negative (DN) T cells. **C** Projection of a selection of protein markers identifying T-cell states. Cells are colored based on the normalized protein expression. **D** Dot plot of the expression of marker genes in the 30 cell clusters. **E** Heatmap showing the Pearson correlation coefficient and its associated *p*-value of cell subset proportions from the 20 CLL LN samples (excl. 2 duplicates), corresponding to the worst performing subset (for which the *p*-value was the highest) of all leave-one-out patient sample sets. Source data are provided as a Source Data file Fig1.

value < 0.05). In addition, we observed a significant correlation between CD4 $T_{CM3}$ cells and time to treatment, as well as CD4 $T_{FH}$ cells and survival. A Kaplan–Meier-survival analysis showed a significant benefit for patients with a higher abundance of CD8 $T_{EM}$ GZMK cells (Supplementary Fig. 4G), suggesting their involvement in CLL control. As 3 of the patients harbored a deletion in 17p, including the *TP53* locus, we specifically compared those cases with TP53-proficient samples. Even though the LN samples of these cases were not distinct, all 3 PB samples with del17p (BC1PB, BC12PB, and BC15PB) clustered together due to a higher abundance of CD4 $T_{EM}$ CD39 cells (Supplementary Fig. 4A, orange cluster). In addition, the del17p samples contained significantly more CD4 $T_{EM}$ GZMK, $T_{PR}$, and DN $T_{EM}$ HELIOS cells compared to TP53 wild-type cases (Supplementary Fig. 4H). This is in line with emerging knowledge that TP53 modulates tumor immunity[20].

## Definition of T-cell exhaustion states and their trajectories in CLL LNs by scRNA-seq

Next, single-cell RNA-seq of T cells from paired CLL LN and PB samples was performed to characterize their transcriptional profile and clonal diversity. CD3[+] T cells and CLL cells (as small spike-in population) of these samples were FACS-sorted (gating strategy shown in Supplementary Fig. 5A) and subjected to scRNA-seq using the 10X Genomics platform, yielding 61,040 cells with an average of 4780 reads per cell and a median of 1807 genes per cell (Supplementary Data 5). Using unsupervised graph-based clustering, we identified 16 clusters presented as a UMAP plot (Fig. 4A), for which we assigned an identity based on the differential gene expression of each cluster compared to the rest of the cells (Fig. 4B). All cell clusters were shared among patients although at different proportions (Supplementary Fig. 5B, C). We defined two clusters consisting of CLL cells, 6 CD4[+] T-cell clusters, 5 CD8[+] T-cell clusters, and a KI-67[+] proliferating T-cell cluster ($T_{PR}$) which contained both CD4[+] and CD8[+] T cells (Fig. 4A, B). More specifically, both CD4[+] and CD8[+] naïve T cells (CD4 $T_N$ and CD8 $T_N$) were defined based on the expression of marker genes such as *LEF1, CCR7* and *SELL*; central memory T cells were identified by a high expression of *IL7R and LTB* for CD8[+] (CD8 $T_{CM}$), and additionally *CD40LG* for CD4[+] T cells (CD4 $T_{CM1}$ and CD4 $T_{CM2}$). Regulatory CD4[+] T cells (CD4 $T_{REG}$) expressed *FOXP3, IL2RA,* and *IKZF2*; and follicular helper CD4[+] T cells (CD4 $T_{FH}$) were defined by the expression of *CXCR5, CD200,* and *ICOS*. A cluster of CD4[+] T cells expressed memory markers and effector molecule genes such as *PRF1* and *GZMA* and was therefore defined as CD4[+] effector memory cells (CD4 $T_{EM}$). Similarly, a high expression of effector molecule genes like *CCL5, PRF1,* and *GZMA* characterized a CD8[+] effector memory cluster (CD8 $T_{EM}$). Two clusters expressed genes characteristic of NK cells, such as *GNLY, FCGR3A,* and *NKG7*. While one was clearly *CD8A* positive and therefore defined as CD8 $T_{EM}$, the other also contained cells without *CD8A* or *CD4* expression, and was therefore named NK-like, as suggested by the pbmc3k signatures from Hao et al.[21] (Supplementary Fig. 5D). We further identified a CD8[+] T-cell cluster with elevated expression of genes related to exhaustion such as *HAVCR2* (coding for TIM3), *LAG3, ENTPD1* (CD39), *PDCD1* (PD1), *CTLA4, TIGIT,* and *TOX* which we defined as CD8 $T_{EX}$ cells. In addition, a cluster that shared many features with CD8 $T_{EM}$ cells, but lacked *PRF1*, and showed expression of *GZMK* and exhaustion-related genes (e.g.

EOMES, LAG3) resembled the CD8 $T_{EM}$ GZMK cluster identified by mass cytometry (Fig. 1B), and was defined as a precursor state of exhaustion (CD8 $T_{PEX}$). Finally, we identified a cluster of T cells that expressed genes characteristic of mucosal-associated invariant T (MAIT) cells, including *SLC4A10* and *TRAV1-2*.

The frequency of the main T-cell clusters was comparable and positively correlated between scRNA-seq and mass cytometry in the 6 samples analyzed by both techniques (Supplementary Fig. 5E), underscoring the robustness of our data. The comparison of T-cell subset distribution between LN and PB revealed an enrichment of CD8 $T_{EM}$, NK-like, and MAIT cells in PB, and CD4 $T_{REG}$, and CD8 $T_{EX}$ in LN (Fig. 4C), which is in agreement with the CyTOF data (Fig. 3C).

To gain a better insight into the transcriptional phenotypes of T cells present in the LNs, we performed a clustering analysis of LN T cells separately and defined 13 cell clusters along the same criteria as described above (Supplementary Fig. 6A–D). Using a published exhaustion score of CD8[+] T cells by Zheng et al.[22], we identified the strongest expression of this gene signature in the CD8 $T_{EX}$ cluster (Fig. 4D), confirming its dysfunctional state, while the defined CD8 $T_{PEX}$ cluster presented the highest score of the precursor exhausted signatures from Guo et al.[23] (Fig. 4E) and Andreatta et al.[24] (Supplementary Fig. 6E) suggesting that these cells constitute a precursor state of exhausted CD8[+] T cells.

To delineate the relationship between T-cell states, we performed trajectory analyses of LN CD4[+] and CD8[+] T cells by destiny[25]. When defining CD8 $T_N$ cells as the starting point of the differentiation path, we observed a divarication into two branches, one containing CD8 $T_{EM}$ cells and the other CD8 $T_{PEX}$ cells, from which $T_{EX}$ cells diverged and constituted a terminal state (Fig. 4F, Supplementary Fig. 6F). When inspecting the differentiation path from CD8 $T_N$ to CD8 $T_{EM}$, four clusters of differentially expressed genes (DEGs) were identified along the trajectory. Naïve marker genes (*LEF1, SELL*) were highly expressed in the first cluster, the second included the transcription factor *KLF2* and *IL6R*, the third activation marker genes (*CD69* and *DUSP1*), and the fourth effector molecule genes (*GZMH, NKG7, GZMA, and FAS*) (Fig. 4G). The differentiation path from CD8 $T_N$ to CD8 $T_{PEX}$ and CD8 $T_{EX}$ was also comprised of four clusters of DEGs, with naïve marker genes (*SELL, IL7R,* and *CCR7*) expressed at the beginning, followed by genes associated with activation (such as *CD69*) and cytotoxic function (*GZMK*), and genes related to exhaustion (*LAG3, TIGIT, EOMES, CD38* and *HAVCR2*) that were highest at the end of the trajectory (Fig. 4H). This data suggests GZMK[+] CD8[+] T cells as a precursor state of terminally exhausted CD8[+] T cells, in line with previously published data[26]. Together with our CyTOF results that showed an accumulation of the CD8 $T_{EM}$ GZMK precursor state and terminally exhausted CD8[+] T cells in CLL LNs (Fig. 3D), these findings suggest cancer-driven exhaustion of CD8[+] T cells in the lymphatic tissue of patients with CLL.

The CD4[+] T-cell trajectory followed a similar pattern as CD8[+] T cells, starting with naïve genes (*SELL, TCF7,* and *IL7R*), followed by genes for activation and effector molecules (*CD44, CXCR3,* and *GZMM*), and then by exhaustion-related genes (*LAG3, EOMES, LAYN, BTLA,* and *IRF4*) (Fig. 4I, J; Supplementary Fig. 6G), indicating cancer-driven exhaustion also of CD4[+] T cells. The CD4 $T_{FH}$ and CD4 $T_{REG}$ cell clusters were diverted into two branches of the trajectory at an early stage (Fig. 4I).

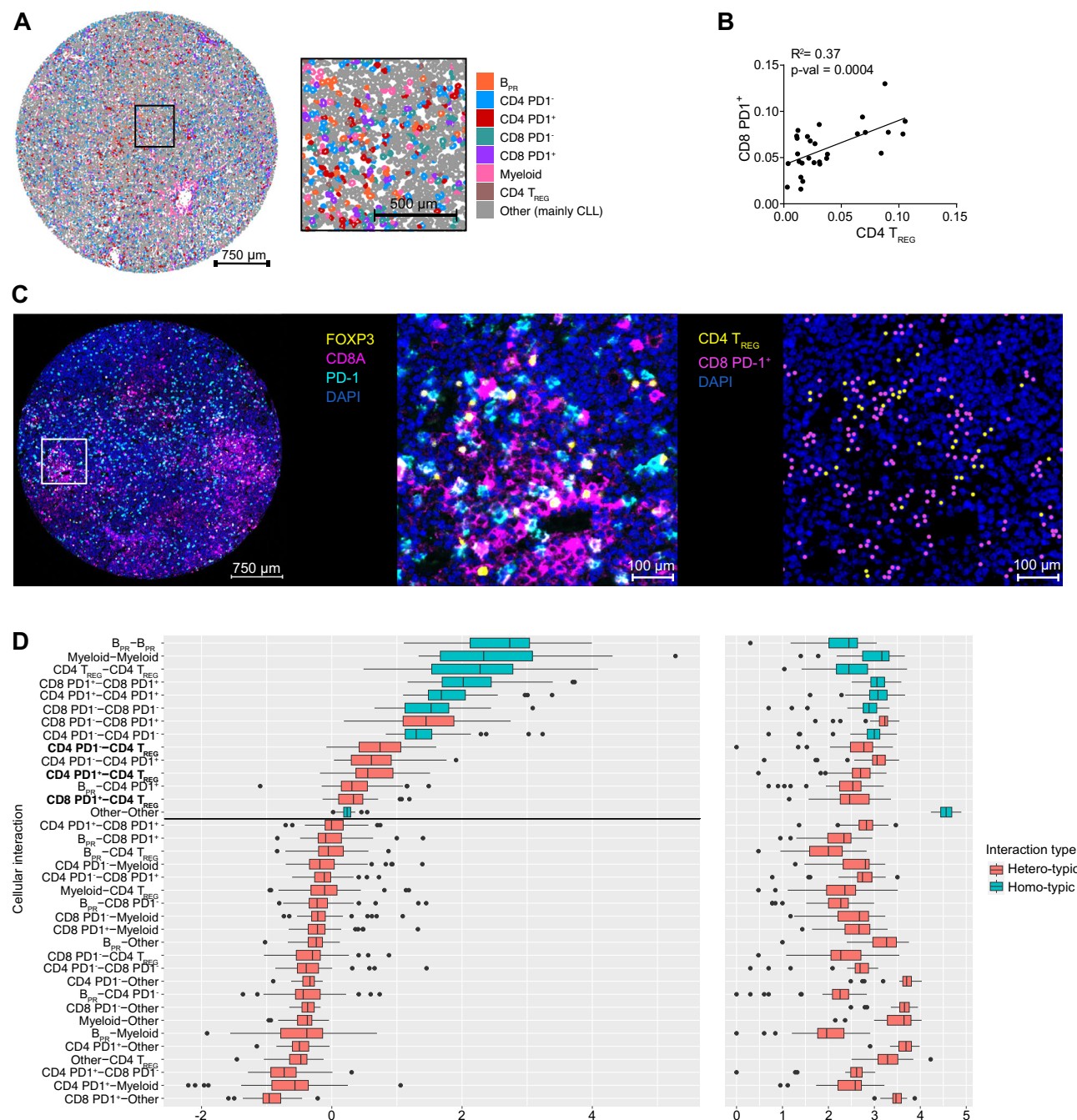

**Fig. 2 | PD1+ T cells reside in close proximity to CD4 TREG cells in CLL LNs.**
**A** Representative CLL LN-derived tissue core stained with 15-color Orion multiplex with cells colored according to subset identification indicated in the legend (a total of 42 LN cores from 29 CLL patients were stained). CLL cells are displayed as $B_{PR}$: Ki-67$^+$ proliferating B cells, and Other: mainly non-proliferating B cells. **B** Pearson correlations of PD1$^+$ CD8$^+$ T cells and CD4$^+$ $T_{REG}$ frequencies determined by multiplex staining of 42 CLL LN cores. Each dot represents an individual patient.
**C** Representative CLL LN-derived tissue core (left image) and field of view (middle image) displaying FOXP3 (yellow), CD8A (pink), PD-1 (light blue), and DAPI (blue) staining (a total of 42 LN cores from 29 CLL patients were stained). The right image

displays cells identified as CD4 $T_{REG}$ and CD8 PD-1$^+$ T cells in yellow and purple dots, respectively. **D** Boxplots (left) depicting the range of enrichment of pairwise interacting cell types for all cores ($n = 42$). Boxplots (right) showing the absolute number of interactions in log10 scale between pairwise interacting cell types across all cores. Boxplots depict the median, the first and third quartile and whiskers extend until 1.5*IQR. Any points beyond the whiskers are outliers and plotted individually. Boxplots are colored by type of interaction, between the same cell type (homotypic, blue) or different cell types (heterotypic, red). Enriched hetero-typic interactions with CD4 $T_{REG}$ are marked in bold. Source data are provided as a Source Data file Fig2.

## TCR analyses reveal differential clonal expansion and CLL-reactivity of T cells in LN compared to PB

The reconstruction of the T-cell receptor (TCR) sequences at the single-cell level allowed us to examine the clonal diversity of T cells within the identified subsets. T cells that shared the same TCR

sequence with at least one other cell were defined as clonal, and clone sizes were categorized in small (2–5 cells), medium (6–20 cells), large (21–100 cells), and hyperexpanded (101–1250 cells). Thereby, we identified a major clonal expansion of CD8$^+$ T cells (ranging from 12% to 76%) and a lower expansion rate of CD4$^+$ T cells

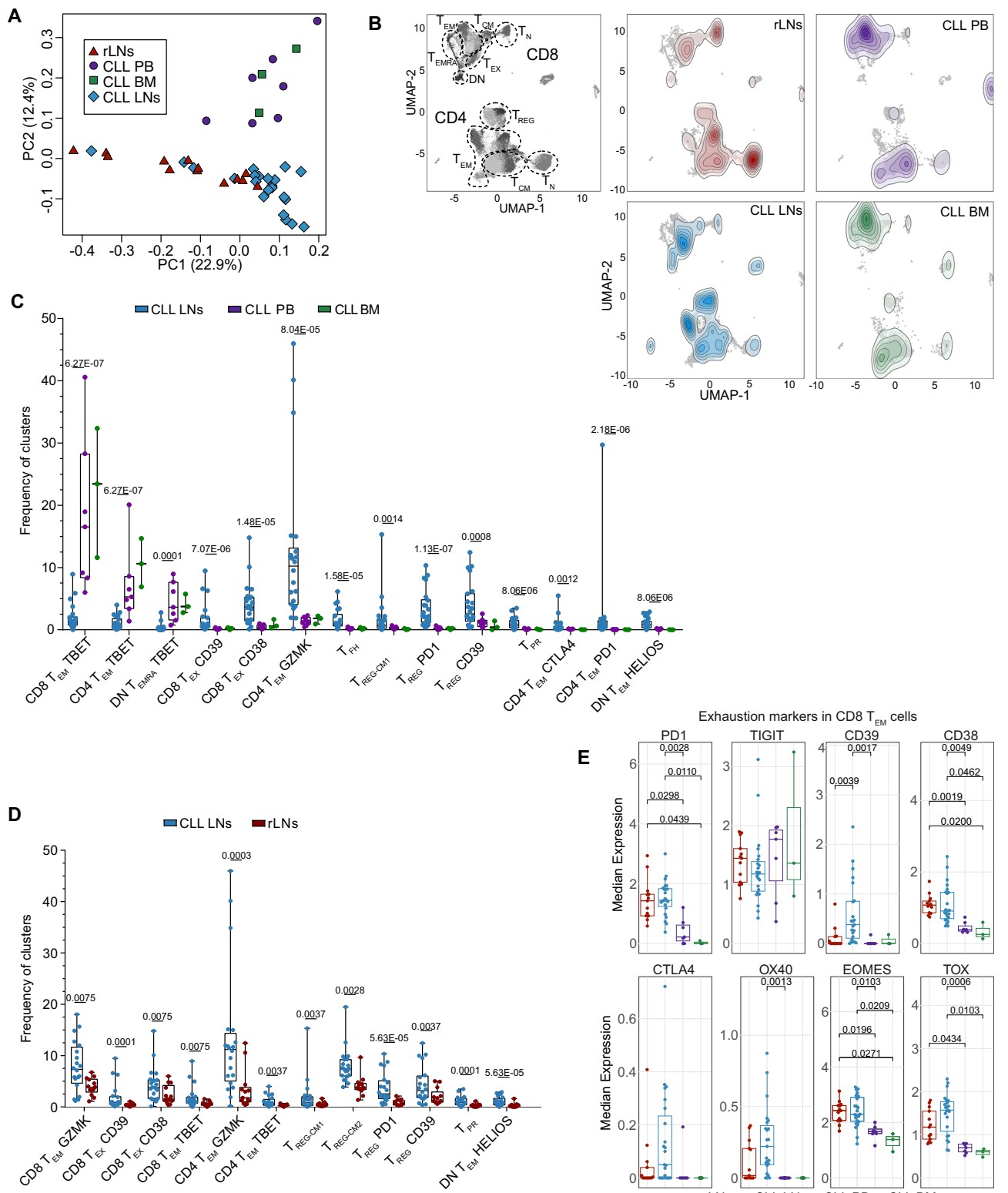

**Fig. 3 | The T-cell composition of CLL LNs is distinct and enriched in regulatory and exhausted subsets. A** Principal component analysis of all samples analyzed by mass cytometry based on cell subset frequencies. **B** UMAP plots of T cells from rLNs, CLL PB, CLL LNs, and CLL BM overlaid with a contour plot indicating the cell density. A UMAP plot indicating the main T-cell clusters is provided on the left. **C** Boxplot showing cell subset abundances out of total T cells in LNs ($n = 20$), PB ($n = 7$), and BM ($n = 3$) of CLL patients. **D** Boxplot showing cell subset abundances out of total T cells in CLL LNs ($n = 20$) and rLNs ($n = 13$). **E** Median expression of PD1, TIGIT, CD39, CD38, CTLA4, OX40, EOMES, and TOX markers in CD8 $T_{EM}$ cells per sample in LNs ($n = 20$), PB ($n = 7$), BM ($n = 3$) samples of CLL patients and rLN samples ($n = 13$) of healthy individuals. Boxplots represent the 25th to 75th percentiles with the median as the central line, whiskers indicate minimal and maximal value. Each symbol represents an individual patient sample. Statistical significance was tested using limma on normalized cell counts, with p-values adjusted for multiple comparisons using the Benjamini−Hochberg method (**C**, **D**), or the Kruskal−Wallis test with Bonferroni correction (**E**). Only significant p-values ($p < 0.05$) are shown. Source data are provided as a Source Data file Fig3.

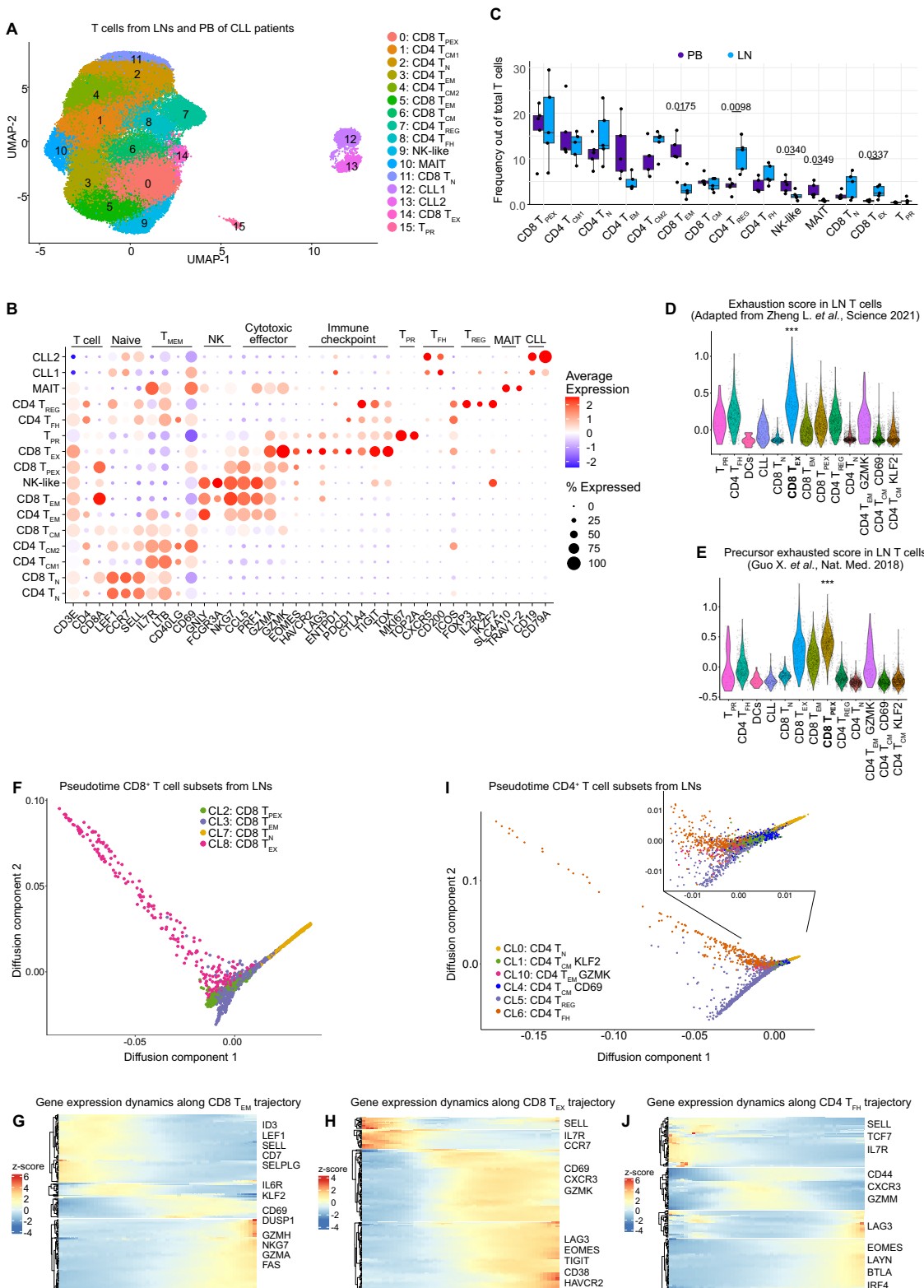

(2% to 38%; Fig. 5A). The expansion of T cells was significantly higher in PB compared to LN samples (Supplementary Fig. 7A), and we did not observe a correlation between the expansion rate of CD4$^+$ and CD8$^+$ T cells. The clonal composition of CD8$^+$ T cells in the LNs was highly variable among patients, with most samples showing predominantly small and medium-sized clones. Notably, T-cell clonal expansion was enriched in the CD8 T$_{EM}$, cluster, followed by CD4

T$_{EM}$, NK-like, and CD8 T$_{PEX}$ subsets in PB, while in the LNs the expansion was mostly in CD8 T$_{EM}$ cluster followed by CD8 T$_{PEX}$ and NK-like cells (Fig. 5B, C, Supplementary Fig. 7B). The cluster distribution of the shared expanded T-cell clones between PB and LNs did not substantially vary, suggesting that clonally expanding T cells maintain the transcriptional phenotype in PB and LNs (Supplementary Fig. 7C).

**Fig. 4 | Single-cell RNA-seq defines T-cell states in CLL LNs. A** UMAP plot of 61,040 cells from paired LNs and PB samples of 5 CLL patients analyzed by scRNA-seq identifying 16 clusters, 6 CD4$^+$ T-cell clusters, 5 CD8$^+$ T-cell clusters, 1 cluster of proliferating CD4$^+$ and CD8$^+$ T cells, 1 cluster of MAIT cells, 1 cluster of NK-like cells, and 2 clusters of CLL cells which were spiked in. **B** Dot plot of the expression of marker genes in the 16 cell clusters. **C** Frequency of cell subset out of total T cells, from PB ($n = 5$) and LN ($n = 5$) samples of CLL patients. A box plot represents the 25th to 75th percentiles and the mean, with dots corresponding to samples. **D-I)** LN samples were clustered and analyzed separately, identifying 13 clusters of T cells and CLL cells (see Supplementary Fig. 6A, B). **D, E** Violin plot of average expression levels in LN T-cell subsets of the slightly adapted exhaustion gene signature derived from Zheng et al.[22] (**D**), and the precursor exhaustion gene signature derived from Guo et al.[23] (**E**). Stars indicate that the CD8 T$_{EX}$ (**D**) and CD8 T$_{PEX}$ (**E**) subsets have statistically significantly higher signature scores compared to all other subsets (see Supplementary Data 5). **F** Pseudotime trajectory across the 4 CD8 T-cell subsets identified in LNs. **G, H** Heatmap showing genes with significant expression changes along the trajectory from CD8 T$_N$ to CD8 T$_{EM}$ (**G**), and from CD8 T$_N$ to CD8 T$_{EX}$ (**H**). Color represents z-scores. **I** Pseudotime trajectory across the 6 CD4$^+$ T-cell subsets identified in LNs. **J** Heatmap showing genes with significant expression changes along the trajectory from CD4 T$_N$ to CD4 T$_{FH}$. Color represents z-scores. Statistical significance was tested by two-sided unpaired $t$ test (**C**) and two-sided Wilcoxon rank sum test (**D, E**). Only significant $p$-values ($p < 0.05$) are shown. Source data are provided as a Source Data file Fig4.

Next, we assessed the clone dynamics between paired LN and PB samples. For two of the patients (BC1 and BC9), we obtained time-matched LN and PB samples at a treatment-naïve stage. From a third treatment-naïve patient (BC3), the PB sample was collected 31 months after the LN sample. For BC0 and BC2, PB samples were taken 67 and 34 months, respectively after the LN sample and, more importantly, these patients received therapy between the two sampling time points (Supplementary Fig. 7D). The top 10 most abundant T-cell clones in the LNs were mostly detectable also in PB of the time-matched samples in untreated patients (BC1 and BC9), although at different frequencies (Fig. 5D). We further detected the majority of the most abundant clones in the LNs also in the PB taken years later in the patient without treatment (BC3). However, there was much less sharing of T-cell clones between LN and PB in the two patients who were treated between the two sampling times (BC0 and BC2), which suggests an expansion of newly arising T-cell clones induced by therapy.

As clonal expansion alone cannot be used as a proxy for tumor reactivity, especially in lymphoid tissues, we used the predicTCR[27] machine learning classifier to identify potentially CLL-reactive T cells in our samples. A percentage of CLL-reactive cells was predicted in all analyzed cases, with a generally higher frequency in LN versus PB samples, and an accumulation of these cells in CD8 T$_{EX}$, T$_{PR}$, CD4 T$_{FH}$, and CD4 T$_{REG}$ clusters, both in LN and PB samples (Fig. 5E, Supplementary Fig. 7E, F). The predicted CLL-reactive cells were predominantly non-expanded, but some clones of NK-like, CD8 T$_{PEX}$, and CD8 T$_{EM}$ clusters were identified (Supplementary Fig. 7G, H). A comparative analysis of T-cell clones showed that the majority of highly expanded clones that are shared between LN and PB samples were predicted to be non-reactive (Fig. 5F). In contrast, predicted CLL-reactive clones were small and resided either only in the LNs or were detectable in LN and PB samples. T-cell clones that were detected only in PB were mostly predicted as non-reactive. Altogether, this suggests that CLL-reactivity of T cells is predominantly happening in LNs with a low circulation rate of these cells to PB.

The presence of convergently recombined TCRs - those with similar CDR3 sequences - has been associated with antigen reactivity in prior studies[28,29]. However, identifying such patterns requires sufficiently deep TCR repertoire sequencing for each patient. To explore the utility of predicTCR as a TCR prioritization tool, we performed bulk TCR repertoire sequencing of T cells from PB samples of the same 5 CLL patients included in the scRNA-seq cohort. Using GLIPH2, we identified clusters of convergently recombined TCRs in each patient (Supplementary Fig. 8). Within these clusters, some TCRs were prioritized by predicTCR as potentially tumor-reactive (Fig. 5G, left)[27], indicating that these may represent candidate CLL-reactive TCRs. However, we emphasize that these predictions are hypothesis-generating and not functionally validated. We also observed clusters of TCRs predicted to be non-tumor reactive.

To further contextualize the repertoire, we performed in silico HLA typing using arcasHLA and cross-referenced identified TCRs with the VDJdb and ImmuneDETECT databases to determine whether any TCR/HLA combinations had been functionally annotated

(Supplementary Data 6). For example, in patient BC9, we identified a GLIPH2 cluster (SP%RNTE_ANQS) with known specificity for the HLA-B*07-restricted CMV epitope RPHERNGFTVL (Fig. 5G, middle)[30], consistent with the patient's HLA type and the known susceptibility of CLL patients to CMV reactivation. This finding highlights the ability of predicTCR to help differentiate virus-reactive from potentially tumor-reactive TCRs, though functional validation was not performed.

We also observed heterogeneous clusters, which we attribute partly to the conservative thresholds used in predicTCR classification and, in some cases, to the mixing of CD4$^+$ and CD8$^+$ T cells within clusters, likely reflecting shared TCR motifs but divergent antigen specificities (Fig. 5G, right). Taken together, these analyses illustrate how predicTCR, by integrating transcriptomic and VDJ sequence data, can assist in prioritizing candidate TCRs for future validation studies, while underscoring the current limitations of relying solely on computational inference to define antigen specificity.

### Prediction of the CLL-T-cell interactome suggests galectin-9 and TIM3 as ligand–receptor pair

Next, we predicted receptor-ligand interactions between the different cell types in the CLL LNs by using the public repository of ligand–receptor interactions from OmniPath[31] to perform differential interaction analysis using CellChat[32]. This identified CLL cells to be highest for predicted outgoing interactions, and CD8 T$_{PEX}$, CD8 T$_N$, CD8 T$_{EM}$ and CD8 T$_{EX}$ for incoming interactions (Fig. 6A, B). As a prominent outgoing signal of CLL cells, MIF was predicted to interact with all antigen-experienced CD8 T-cell clusters via CD74, CXCR4 and CD44 (Fig. 6C), which is of interest as MIF was identified as a driver of CLL development in the Eμ-TCL1 mouse model[33]. MIF was also among the most prominent outgoing signals predicted for CD4 T$_{REG}$ (Supplementary Fig. 9A) which in general strongly mirrored signals provided by CLL cells. We further identified galectin-9 (encoded by *LGALS9*) as a prominent signal from CLL cells to all other cell types, including a cell type-specific interaction with TIM3 (encoded by *HAVCR2*) on CD8 T$_{EX}$ (Fig. 6C). To estimate which of the identified interactions are specific for CLL and not common to non-malignant lymph nodes, we included published scRNA-seq data from 5 rLNs in this analysis (Supplementary Fig. 9B, C)[34]. T-cell subsets in this published and our datasets were grouped into major functional types: CD4 T$_N$, CD8 T$_N$, CD4 T$_{FH}$, CD4 T$_{CM}$ (including CD4 T$_H$ KLF2 and CD4 T$_H$ CD69), CD4 T$_{REG}$, CD4 T$_{EM}$, and CD8 T$_{EM}$ (including CD8 T$_{EM}$, CD8 T$_{PEX}$, and CD8 T$_{EX}$) to facilitate comparison. Predicted receptor-ligand interactions were prioritized by the probability of differential interactions across CLL LNs and rLNs. Overall, the number of inferred interactions was higher in CLL LNs compared to rLNs (1828 vs. 1622; Supplementary Fig. 9D). The highest number of predicted interactions was found for CLL cells and CD8 T$_{EM}$ cells, which act as senders and receivers of signals, both in an autocrine and paracrine manner (Fig. 6D, E). These two cell types were predicted to strongly interact with each other and also with all other cell subsets. Galectin-9 and MIF remained relevant outgoing signals from CLL cells also in this differential analysis (Fig. 6F). In addition, several

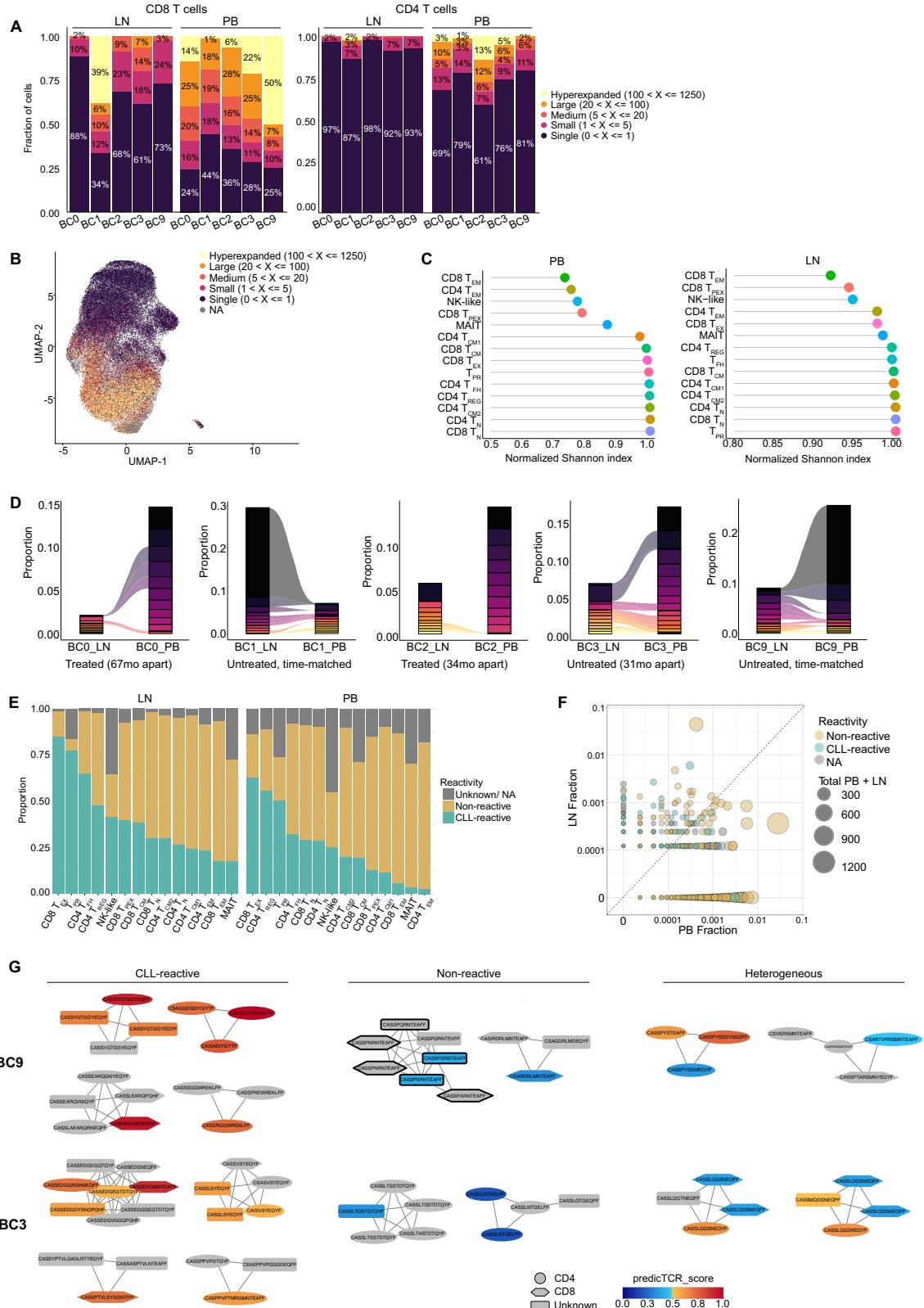

cytokines and chemokines with known relevance in CLL (TGFβ, CCL4, CCL5, IL16, IFNγ)[35,36], as well as collagen and adhesion molecules (COL9A2/3 and ICAM-ITGAL) were predicted to be part of CLL-specific interactions (Fig. 6F). Numerous inferred interactions between T cells and CLL cells were related to T-cell inhibitory signals, such as HLA-LAG3[37], BTLA-CD247[38], *ENTPD1-ADORA2A*[39], and *LGALS9*-related circuits, suggesting their relevance in immune escape. Of

interest, galectin-9 levels are increased in serum of patients with CLL[40–42], and binding of galectin-9 to TIM3 induces T-cell death and thus contributes to tumor immune escape[43,44]. Our scRNA-seq data identified CLL cells as a source of galectin-9, and an increased *LGALS9* expression in dendritic cells in CLL compared to rLNs (Supplementary Fig. 9E). The galectin-9 binding partner TIM3 (encoded by *HAVCR2*) was detected in CD8 T$_{EX}$ cells in CLL LNs, but was absent in

**Fig. 5 | TCR analyses reveal increased tumor-reactive T cells in the LNs. A** Bar plot indicating the percentage (rounded values are indicated) of single, small, medium, large and hyperexpanded-sized clones in CD8+ (left) and CD4+ (right) T cells in LN and PB for each patient analyzed (*n* = 10). **B** UMAP plot colored according to the T-cell clone size based on the TCR-seq data. NA: no TCR information available. **C** Graph showing the TCR Shannon diversity index for each T-cell subset identified by scRNA-seq in PB and LN samples. The dot color corresponds to the UMAP cluster plot from Fig. 4A. **D** Alluvial plot displaying the top 10 most frequent clones for LN and PB. **E** Proportion of predicted CLL-reactive, non-reactive and unknown/ NA T-cell clonotypes out of total T cells in LN and PB. **F** Scatter plot shows LN and PB clone sizes from all 5 CLL patients. Color represents reactivity status and dot size the total number of cells per clonotype. **G** Left: Examples of large clusters of convergently recombined TCRs identified by GLIPH2 containing multiple CLL T-cell-derived TCRs predicted to be CLL-reactive (orange-red), as well as TCRs found in the LN or PB for which no scSEQ data and predicTCR scores were available (grey). Middle: Examples of TCR clusters called as non-CLL reactive (blue); in patient BC9 7 TCRs within the SP%RNTE_ANQS cluster are known to bind the HLA-B*07 restricted epitope of the CMV pp65 protein (bold black node border). Right: Examples of heterogeneous clusters. TCRs for which CD4/CD8 status could not be determined due to lack of scSEQ data are illustrated as rectangular nodes. Source data are provided as a Source Data file Fig5.

rLNs (Supplementary Fig. 9E). Multiplex immunofluorescence staining of CLL LN sections further confirmed the presence of TIM3-expressing CD4+ and CD8+ T cells in the tissue (Supplementary Fig. 9F).

### Galectin-9 blockade reduces CLL development in mice

We next aimed to preclinically test the potential of galectin-9 blockade in a mouse model of CLL. To evaluate the validity of the commonly used Eμ-TCL1 mouse line[45] for this study, we characterized the splenic T-cell compartment of 2 mice after adoptive transfer of TCL1 leukemia (TCL1 AT) by single-cell RNA-seq. This allowed us to identify 13 clusters of T cells, including naïve, effector, effector memory, and regulatory T-cell subsets (Fig. 7A, B, Supplementary Fig. 10A). When integrating TCR-seq data, it became clear that one of the two analyzed samples (#107) harbored a hyperexpanded CD4+ T-cell clone showing an effector phenotype and *Gzmk* expression, which was not the case for the second sample (#110) that contained a large CD8+ effector memory T-cell cluster (Supplementary Fig. 10B, C). The clonally expanded CD4+ effector T cells in sample #107 contained a cluster of cells expressing exhaustion-related genes. This data suggests the occurrence of CD4+ T-cell-driven adaptive immune control in TCL1 AT mice which is in line with observations in other tumor mouse models and patients with cancer[16,46–48].

Flow cytometry analyses showed that the vast proportion of malignant B cells in the spleen and bone marrow of TCL1 AT mice expressed galectin-9 which was not the case for B cells from wild-type (WT) mice (Fig. 7C, Supplementary Fig. 10D). Also, higher frequencies of galectin-9-positive CLL cells were detected in LNs of TCL1 AT mice compared to B cells from WT mice (Supplementary Fig. 10D). We further observed TIM3-positive T cells in tumor-bearing TCL1 AT but not WT mice, with the highest frequencies detected in CD8 $T_{EF}$ and CD4 $T_{REG}$ cells (Fig. 7D). Altogether, we concluded that the TCL1 AT model is useful for testing the potential of galectin-9 as immunotherapy target for CLL.

Therefore, we treated CLL-bearing TCL1 AT mice with galectin-9-blocking antibodies for 20 days (Fig. 7E). By a weekly assessment of CLL cell counts in the blood of the mice, a deceleration of leukemia development was observed by anti-galectin-9 compared to isotype control treatment (Fig. 7F). At the endpoint of the experiment, 7 weeks after TCL1 AT, a reduced spleen weight and lower numbers of CLL cells were detected in the spleen and lymph nodes of the mice receiving anti-galectin-9 (Fig. 7G, Supplementary Fig. 10E). We further analyzed T cells in the spleen and lymph nodes at the endpoint and did not observe major differences in the numbers of CD8+ T cells, conventional CD4+ T cells, or CD4+ $T_{REG}$, nor in their activation state. We detected however a lower frequency of CD39+ and a higher frequency of CD107A+ CD8+ T cells in the treated mice (Fig. 7H, I), suggesting a higher functionality of CD8+ T cells upon anti-galectin-9 treatment. The most drastic differences were observed in the frequencies of TIM3+ T cells, which were drastically decreased for CD8+ T cells, conventional CD4+ T cells, and CD4+ PD1+ $T_{REG}$ cells in the spleen, and to a lower degree also in LNs (Fig. 7J–L, Supplementary Fig. 10F) which is in line with the proposed role of galectin-9 in preventing apoptosis of TIM3+

T cells[44]. Based on these data, we suggest that anti-galectin-9 treatment results in a better immune control of CLL via suspending the immune suppressive activity of TIM3 on CLL-associated T cells.

### High galectin-9 expression is associated with shorter survival of patients with CLL, kidney or brain tumors

Next, we explored the expression of *LGALS9* across cancer entities using TCGA data[49]. A comparison of tumor and respective normal tissue revealed a higher expression in about half of the cancer types analyzed (Fig. 8A, Supplementary Fig. 11A). To estimate the relevance of galectin-9 in CLL development, we analyzed a previously published proteome data set of CLL cells[50]. Dividing patients into two groups based on the median level of galectin-9 expression clearly demonstrated a significantly shorter treatment-free survival in cases with high galectin-9 expression (Fig. 8B). Dividing this cohort of patients into the two main prognostic groups, namely cases with unmutated or mutated *IGHV* gene locus, we observed a generally high heterogeneity in galectin-9 levels in both groups, and a significantly higher expression in the *IGHV*-unmutated group which has a worse overall prognosis (Supplementary Fig. 11B). Within this group, higher galectin-9 protein levels clearly predicted shorter treatment-free survival (Supplementary Fig. 11C) which was not the case in the *IGHV*-mutated group of CLL patients (Supplementary Fig. 11D).

We further analyzed the prognostic value of *LGALS9* expression across cancer types and identified significant associations of high levels of *LGALS9* with shorter overall survival for renal cell carcinoma and glioma patients (Fig. 8C, D). By exploring published scRNA-seq data of these two cancer types, we observed *LGALS9* expression mainly in tumor-associated myeloid cells (Fig. 8E–J). Of interest, myeloid cell infiltration has been linked to immune suppression and worse outcome of patients for renal cell carcinoma and glioma[51–54]. Altogether, this suggests galectin-9 as potential target for immunotherapy in CLL and likely other cancer entities.

## Discussion

T-cell exhaustion is considered one of the main reasons for the failure of immunotherapies, including immune checkpoint inhibitors, CAR-T cells, or bispecific T-cell engagers. However, its role in treatment resistance among patients with CLL, who often exhibit limited therapeutic response to these therapies, has not been explored so far. Our study provides a single-cell-resolved analysis of the T-cell landscape in blood and tissue samples of CLL, the most common leukemia in the Western world. CLL is a disease that arises in the lymph nodes, but the malignant B cells accumulate also in the peripheral blood. Most previous studies characterized disease-associated alterations in the T-cell compartment in blood samples. Our findings reveal that exhaustion of tumor-reactive cytotoxic T cells and accumulation of regulatory T cells are very prominent in the lymph nodes but not in blood, suggesting the lymphoid tissue as the site of cancer-directed immune responses in CLL. This in-depth characterization of the T-cell landscape in CLL LNs suggests that failure to immunotherapies can be attributed to CLL-induced T-cell exhaustion, confounded by an immunosuppressive microenvironment enriched with $T_{REG}$ cells.

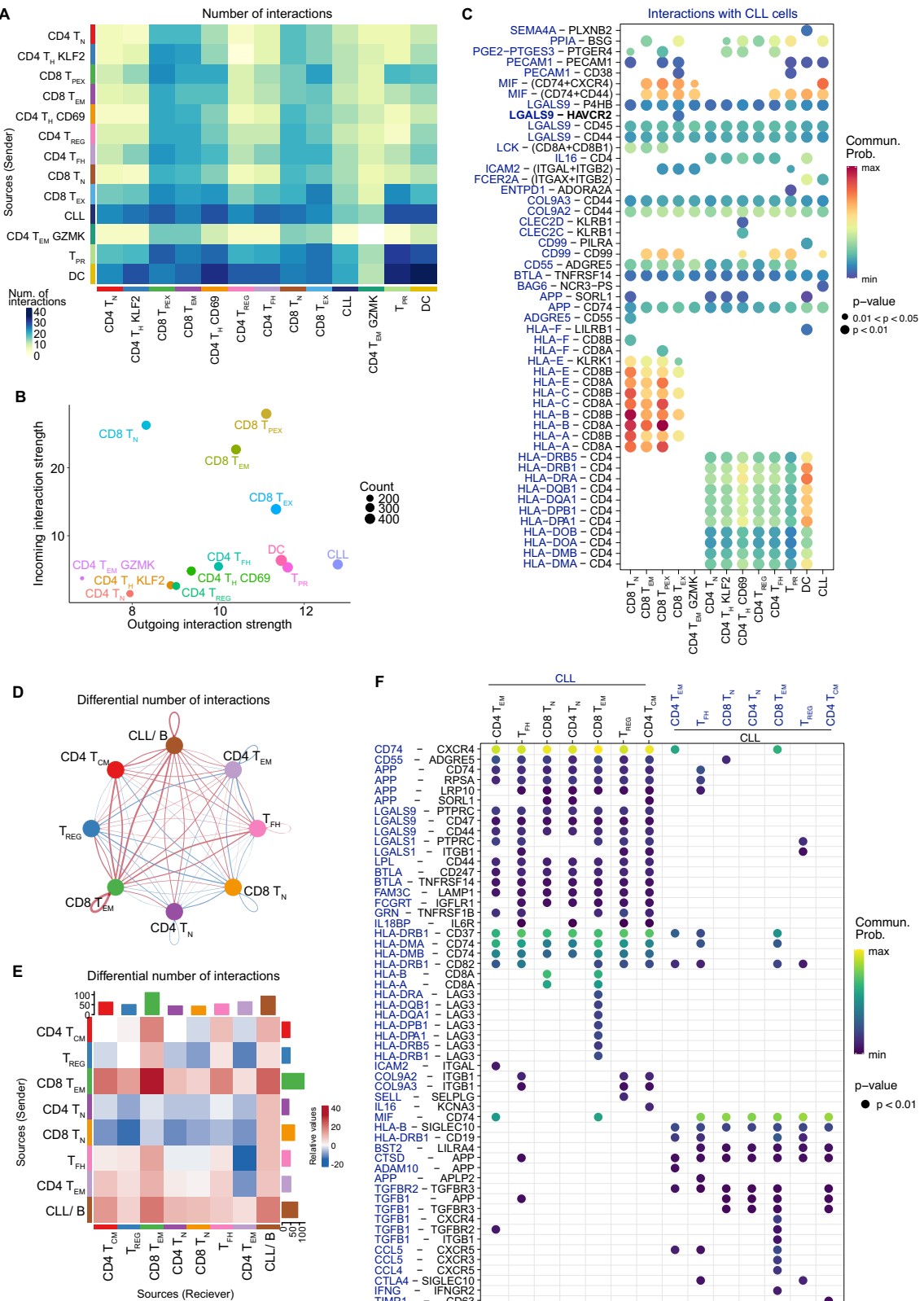

Using the predicTCR and GLIPH2 tools, we identified clusters of TCRs with features suggestive of shared antigen specificity, which may include CLL-associated reactivities. However, we stress that these results are exploratory and do not constitute direct evidence of CLL-specific T-cell responses. The antigenic targets of these TCRs remain unknown, and while previous studies have suggested that genomic aberrations or the leukemic immunoglobulin genes could serve as potential neoantigens in CLL[55,56], further investigation is required to substantiate these hypotheses. Moving forward, cloning the herein-predicted CLL-reactive TCRs and screening their reactivity against autologous CLL cells will be critical for validating these observations and further advancing our understanding of the mechanisms driving immune dysfunction in CLL, with the final aim to improve effective antigen-specific immunotherapies. In this study, predicTCR was used

**Fig. 6 | Interactome analyses predict a robust and disease-specific cross-talk of CLL and T cells in LNs including galectin-9 circuits. A–C** Cell-cell communication network was analyzed on scRNA-seq data from 5 CLL LNs using CellChat. **A** Heatmap depicting the number of interactions between cell subsets in CLL LNs. **B** Scatter plot showing the dominant sender (*X*-axis) and receiver (*Y*-axis) cell subsets. **C** Heatmap plot depicting the list of significant ligand–receptor pairs between CLL cells (molecule in blue) and all the other cell subsets (molecule in black). The dot color and size represent the calculated communication probability and *p*-values, which are computed from one-sided permutation test. **D–F** Differential cell-cell communication networks between CLL LNs and rLNs were analyzed using CellChat. **D** Circle plot depicting the differential number of interactions between cell subsets in CLL LNs compared to rLNs. Thickness of bands represents the number of differential interactions between the two data sets, and increased interactions are depicted in red, decreased interactions in blue. **E** Heatmap plot showing the differential number of interactions of CLL LNs versus rLNs. Rows and columns represent cell subsets acting as sender and receiver, respectively. Bar plots represent the total outgoing (right) and incoming (top) interaction scores, respectively, for each cell subset. **F** Heatmap plot depicting a curated list of ligand–receptor pairs differentially upregulated in CLL LNs compared to rLNs as identified via CellChat[32]. The dot color and size represent the calculated communication probability and *p*-values of differential communication, respectively. Significantly differentially upregulated ligand–receptor pairs were calculated via the Wilcoxon rank-sum test. The first 7 columns show interactions sent by CLL cells (molecule on CLL cells in blue), while the last 7 columns show interactions received by CLL cells (molecule on CLL cells in black). Source data are provided as a Source Data file Fig6.

as a prioritization tool to identify candidate TCRs for downstream validation, not as a means of confirming tumor reactivity.

In addition to terminally exhausted CD8+ T cells that are enriched in the lymph nodes, a population of cells that resemble the previously described precursor cells of exhaustion ($T_{PEX}$)[8,9], that share features with memory T cells, and have a high *GZMK* expression[26], were identified in CLL PB and LN samples. Our data identifying such cells in CLL LNs is in line with recent studies demonstrating their presence in tumor-draining LNs of solid cancer, where they are essential for efficient response to ICI therapy[57,58]. A recent study identified an intermediate exhausted CD8 effector/effector memory T-cell cluster marked by *ZNF683* expression in bone marrow samples from patients with Richter's transformation (RT), which correlates with anti-PD1 therapy response[59]. In our scRNA-seq analysis of lymph node samples, we found two small *ZNF683*-expressing cell clusters in CD8 $T_{EM}$ and CD4 $T_{FH}$ cells, but none in the CD8 $T_{PEX}$ cluster. Even though this comparison is limited by looking at T cells from different tissues, this suggests potential differences in T-cell exhaustion states between CLL and RT, possibly contributing to the efficacy of ICI therapy which is higher in RT.

CLL cells in lymph nodes are densely packed together with T cells and interactome analyses using the scRNA-seq data suggested a list of molecular interactions between the two cell types. It is well-accepted that CLL cells depend on microenvironment-derived signals for survival and proliferation, and the identified interactions including MIF-CD74[33], CCL4 and CCL5[36], and INFγ[60] are part of this support. Further, the interactome study inferred several inhibitory molecular interactions including BTLA-CD247, CTLA4, and galectin-9 (encoded by *LGALS9*). Galectin-9 is a known ligand of TIM3, an inhibitory receptor that is expressed on exhausted T cells[61]. And even though we detected very low levels of *HAVCR2* (encoding TIM3) by scRNA-seq, we confirmed the presence of TIM3-positive T cells in CLL LN tissue, and assumed an involvement of this interaction in immune escape as galectin-9 binding to TIM3 promotes CD4+ $T_{REG}$ development in CLL[62].

TIM3-targeting antibodies now enter clinical trials and show promising anti-tumor activity in patients with advanced solid tumors[63]. These antibodies block the interaction of TIM3 with phosphatidylserine and CEACAM1, but only partially the binding to galectin-9[64,65]. Blocking antibodies targeting galectin-9 can overcome this limitation and suppress tumor growth in combination with chemotherapy in a breast cancer mouse model[66], and in ex vivo models of follicular lymphoma[67]. This activity was dependent on CD8+ T cells, which is in line with our observation of an enhanced CD8+ effector function in anti-galectin-9 treated mice with TCL1 leukemia.

The binding of galectin-9 to TIM3 on T cells induces cell death thereby limiting adaptive immunity[44]. This raises the question of why terminally exhausted T cells harboring TIM3 expression persist in the tumor microenvironment. Recent data showed that PD1, via physically interacting with galectin-9 and TIM3, protects T cells from apoptosis[68]. As a consequence, PD1+ TIM3+ T cells accumulate in tumors even in the presence of enhanced levels of galectin-9. Blockade of galectin-9

in vitro and in mouse models results in enhanced T-cell survival and improved anti-tumor immunity, especially in combination with co-stimulatory anti-GITR treatment[68]. In accordance, anti-galectin-9 treatment of TCL1 AT mice diminished PD1+ TIM3+ T cells along with a reduction in leukemia development. Besides improving T-cell responses, ligand activation of galectin-9 in the tumor microenvironment of pancreatic ductal adenocarcinoma results in tolerogenic macrophage programming and adaptive immune suppression[69]. As CLL is dominated by immunosuppressive myeloid cells[70], galectin-9 blockade potentially ameliorates this milieu which likely contributes to a better T-cell function.

In line with these mechanistic findings, *LGALS9* expression is higher in tumor compared to respective normal tissue across many cancer types. In CLL, we defined the malignant B cells as the main source of galectin-9, even though myeloid cell types likely contribute to its expression. In solid cancers however, galectin-9 is likely produced mainly by myeloid cells, like tumor-associated macrophages or myeloid-derived suppressor cells. This also explains that both the abundance of tumor-infiltrating myeloid cells and high expression of *LGALS9* predict shorter survival of patients with renal cell carcinoma or glioma. Prediction of outcome is restricted in CLL to cases with unmutated *IGHV* gene locus, which show significantly higher galectin-9 expression compared to cases with mutated *IGHV* and are generally considered as the worse prognostic group of patients[71,72]. As the Eμ-TCL1 mouse model mimics CLL with unmutated immunoglobulins[73], our data suggest treatment efficacy of anti-galectin-9 specifically for this more aggressive form of CLL.

Our study focused on an in-depth characterization of the T-cell landscape in CLL and we excluded myeloid cells from the analysis, which represents a limitation of our analysis. Immunosuppressive interactions of CLL-associated myeloid cells and T cells contribute to immune escape in CLL[74,75], but their analysis requires fresh tissue samples to prevent a biased loss of myeloid cell subsets by freezing and thawing, and such samples are rarely available. As suppression of anti-tumor immunity is complex and multifactorial, obtaining a complete picture of all cell types and their interactions with the cancer cells will be necessary for the development of combinatorial treatment targeting multiple immune cell types and checkpoints to avoid immune escape and resistance to therapy.

## Methods

### Ethics declarations

Lymph node (LN, *n* = 22), peripheral blood (PB, *n* = 7), and bone marrow (BM, *n* = 3) samples from CLL patients, and reactive lymph node (rLN, *n* = 13) samples from heathy controls were obtained after informed consent and according to the guidelines of the Ethics Committees of the medical faculties of the University of Barcelona and the University of Heidelberg, and the Declaration of Helsinki. Patients with CLL were diagnosed following the World Health Organization (WHO) classification criteria. All clinical information of the patients analyzed in this work is provided in Supplementary Data 1. The

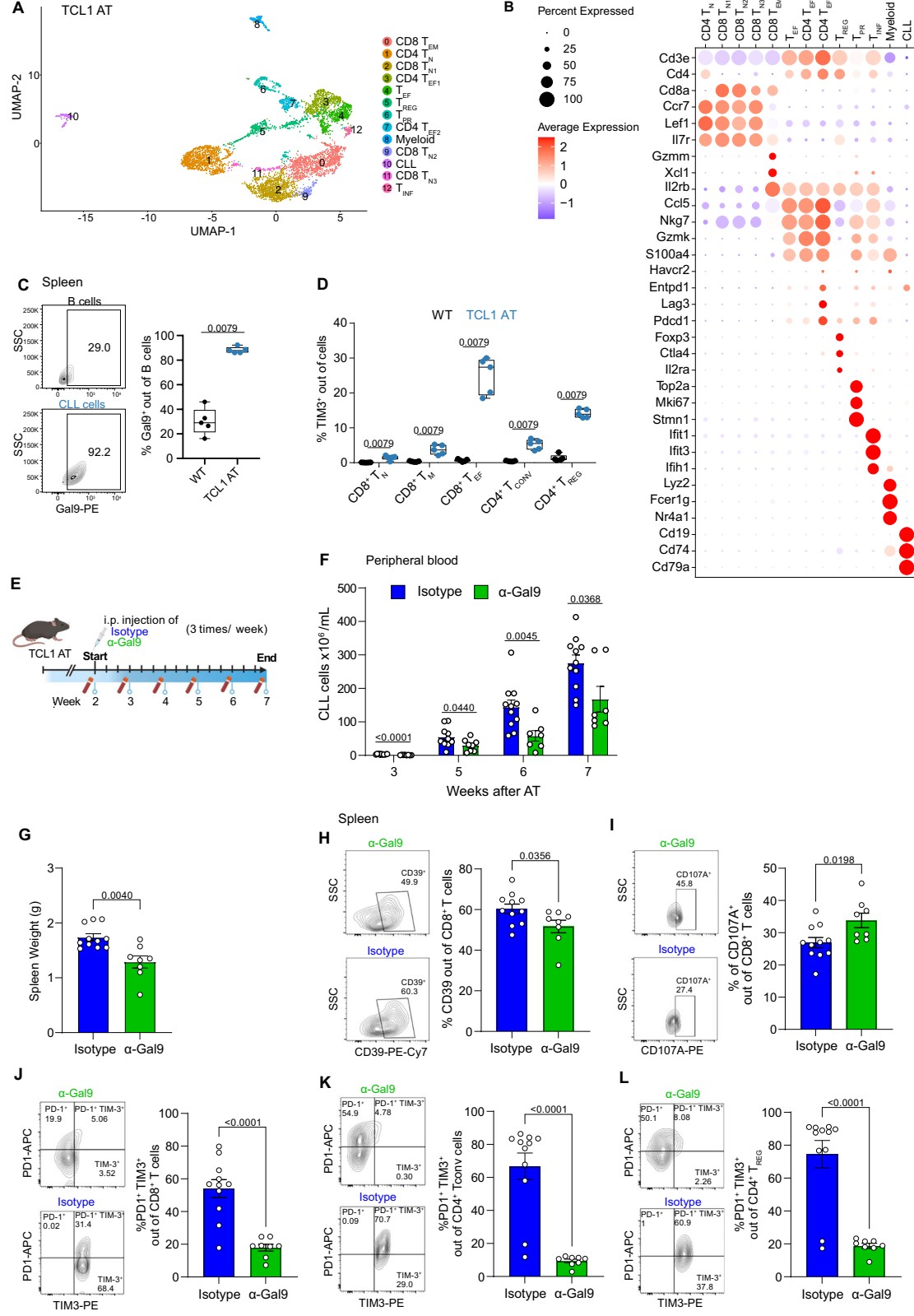

provided information does not allow for the identification of the individuals.

All experiments involving laboratory animals were conducted in pathogen-free animal facilities at the German Cancer Research Center in Heidelberg or the Luxembourg Institute of Health in Luxembourg with the approval of the Regierungspräsidium Karlsruhe (G-77/19 and G-112/21) and the Luxembourg Ministry for Agriculture (#LUPA 2019/21), respectively. Mice were treated following the European guidelines.

## Mice and tumor models

Female C57BL/6 wild-type (WT) mice were purchased from Charles River Laboratories (Germany) or Janvier Labs (France). Adoptive transfer of TCL1 leukemia cells was performed following our

**Fig. 7 | Blocking of galectin-9 controls tumor growth in the TCL1 mouse model.** **A** UMAP plot of 6,201 cells from the spleens of 2 mice after adoptive transfer of TCL1 leukemia (TCL1 AT) analyzed by scRNA-seq identifying 13 clusters, including 11 T-cell clusters, CLL cells and myeloid cells. **B** Dot plot of the expression of marker genes in the 13 cell clusters. **C** Representative contour plot and percentage of galectin-9$^+$ (Gal9$^+$) B cells and CLL cells from spleen of wild-type control (WT; $n = 5$) and TCL1 AT ($n = 5$) mice, respectively. **D** Percentage of TIM3$^+$ cells out of CD8 T$_N$, CD8 T$_M$ (memory cells), CD8 T$_{EF}$ (effector cells), CD4 T$_{CONV}$ (conventional), and T$_{REG}$ T cells in spleen of WT ($n = 5$) and TCL1 AT ($n = 5$) mice measured by flow cytometry. **E** Schematic diagram of treatment of TCL1 AT mice with galectin-9-blocking antibody (α-Gal9). Analyses of T, myeloid and CLL cells were performed 7 weeks after treatment start in isotype antibody- ($n = 11$) and α-Gal9-treated ($n = 8$) mice. Created in BioRender. Floerchinger, A. (2025) https://BioRender.com/x4or7yz. **F** Absolute number of CD19$^+$ CD5$^+$ CLL cells in blood of isotype antibody- ($n = 11$) and α-Gal9-treated ($n = 8$) mice. **G** Spleen weight of isotype antibody- ($n = 11$) and α-Gal9-treated ($n = 8$) mice. **H** Representative contour plot and percentage of CD39$^+$ cells out of CD8$^+$ T cells from spleen of isotype antibody- ($n = 11$) and α-Gal9-treated ($n = 8$) mice. **I** Representative contour plot and percentage of CD107A$^+$ out of CD8$^+$ T cells from spleen of isotype antibody- ($n = 11$) and α-Gal9-treated ($n = 8$) mice. **J–L** Representative contour plot and percentage of PD1$^+$ TIM3$^+$ cells out of CD8$^+$ T cells (**J**), CD4$^+$ T$_{CONV}$ (**K**), and T$_{REG}$ (**L**) cells from spleen of isotype antibody- ($n = 11$) and α-Gal9-treated ($n = 8$) mice. Each symbol represents an individual mouse, and statistical significance was tested by two-sided unpaired $t$ test with Welch approximation. Boxplots represent the 25th to 75th percentiles with the median as the central line, whiskers indicate minimal and maximal value (**C**, **D**), bars plots indicate mean ± SEM (**F–L**). Source data are provided as a Source Data file Fig7.

standard protocol[37,74]. In short, malignant B cells were enriched from splenocytes of Eμ-TCL1 mice using EasySep Mouse Pan-B Cell Isolation Kit (StemCell Technologies, Inc., Cologne, Germany). The CD5$^+$CD19$^+$ content of purified cells was typically above 95%, as measured by flow cytometry. $1 \times 10^7$ malignant TCL1 splenocytes were transplanted by intravenous (i.v.) injection into 8-12 weeks-old C57BL/6 WT female mice. Leukemia progression was monitored weekly by assessment of the percentage of CD5$^+$CD19$^+$ cells in the PB. The use of female mice was justified due to a more homogeneous disease development in those compared to male mice, and due to no major differences observed in the T-cell landscape of male and female CLL patients.

### Antibody treatments
Treatment of mice with anti-galectin-9 antibodies was performed at the LIH animal facility with the approval of the Luxembourg Ministry for agriculture and following EU guidelines. Mice were maintained under standard housing conditions: a 12 h light / 12 h dark cycle, ambient temperature of $22 \pm 2$ °C, and relative humidity between 45% and 65%. Two weeks after transplantation with TCL1 leukemia cells, mice were assigned to different treatment arms according to the percentage of CD5$^+$CD19$^+$ (CLL) cells out of live cells in PB to ensure comparable average CLL burden across groups prior to treatment. Subsequently, mice were injected i.p. with 6 mg/kg of anti-galectin-9 (clone: RG9-1) or rat IgG2b isotype control antibody (clone: LTF-2), 3 times per week for another 7 weeks. All antibodies for in vivo experiments were acquired from BioXcell (West Lebanon, NH). The maximal tumor burden permitted was defined as 95% CLL cells and 200 million CLL cells per mL of blood. This limit was not exceeded in any of the experiments. At the endpoint, mice were euthanized by cervical dislocation.

### Tissue sample collection and preparation of cell suspensions
Mice were euthanized by increasing concentrations of carbon dioxide ($CO_2$) or by cervical dislocation before reaching the established humane endpoint. PB was drawn from the submandibular vein or via cardiac puncture and collected in ethylenediaminetetraacetic acid (EDTA)-coated tubes (Sarstedt). Single-cell suspensions from spleens, bone marrow (BM) and inguinal LNs were prepared[74,76]. BM cells were flushed from femurs with 5 mL of phosphate-buffered saline (PBS)/5% fetal calf serum (FCS). Spleen single-cell suspensions were generated by using the gentleMACS tissue dissociator with Gentle MACS tubes C (Miltenyi Biotec). Single-cell suspensions from LNs were prepared by grinding the tissue through 70 μm cell strainers (BD Biosciences). Erythrocytes were lysed by using Red blood cell lysis buffer (Mitenyi Biotec).

### Flow cytometry
Single-cell suspensions were immunostained with antibodies against cell surface proteins (Supplementary Data 2) in FACS buffer containing 0.1% fixable viability dye (Thermo Fisher Scientific, 65-0866-14) for 30 min at 4 °C. Cells were subsequently washed twice in FACS buffer, fixed with IC fixation buffer (Thermo Fisher Scientific, 00-8222-49) for 30 min at room temperature, washed twice with FACS buffer, and stored at 4 °C in dark conditions until being analyzed.

For transcription factor or intracellular cytokine staining, cell surface-stained cells were fixed for 30 min at room temperature with Foxp3 fixation/permeabilization buffer (Thermo Fisher Scientific, 00-5523-00 or Miltenyi Biotec, 130-093-142). After a washing step with FACS buffer, cells were permeabilized for 30 min at room temperature with 1X permeabilization buffer (Thermo Fisher Scientific, 00-8333-56). Intracellular staining with antibodies against transcription factors or cytokines was performed in 1X permeabilization buffer for 30 min at room temperature. Excess antibodies were washed twice with 1X permeabilization buffer and cells were resuspended in 1X permeabilization buffer and stored at 4 °C in dark conditions until they were analyzed by flow cytometry.

For intracellular CD107a assessment, splenocytes were stimulated overnight with phorbol myristate acetate (PMA) and ionomycin (100 nmol/L and 1 μmol/L) and incubated for 4 h with Brefeldin A (BFA, 1X) prior to washing and cell surface staining.

Cell fluorescence was assessed using a BD LSRFortessa (BD Biosciences) or a Novocyte Quanteon (Agilent) flow cytometer, and data were analyzed using FlowJo X 10.0.7 software (FlowJo). In each experiment, single fluorochrome stainings were used to compensate for spectral overlap. Fluorescence minus one (FMO) controls were employed for proper gating of positive cell populations. FMO-normalized Mean Fluorescence Intensity (nMFI) for unimodal distributions or percentage of positive cells for bimodal distributions were determined for populations of interest. The standard gating strategy for analysis of CLL cells and T cells is depicted in Supplementary Fig. 5A.

### Immunofluorescence of whole lymph node sections
Paraffin-embedded tissue sections of lymph nodes were obtained from the National Center for Tumor Disease (NCT) Heidelberg, Hospital Clínic de Barcelona, and the University of Würzburg.

After validation of primary and secondary HRP-conjugated antibodies using DAB chromogenic detection, incubation times and appropriate antibody concentrations were optimized using the Opal 5-color kit (Akoya Biosciences, NEL840001KT). Epitope stability was assessed to determine the order of each antibody in sequence, and antigen stripping efficiency was confirmed in all used primary antibodies.

De-paraffinization and rehydration of lymph node sections were performed by immersion in xylene for 10 min twice, followed by immersion in a series of descending ethanol concentrations prior to distilled water. Slides were then cooked in a steam cooker with antigen retrieval buffer pH9 (Akoya Biosciences, AR900250ML) at 100 °C for 25 min. Slides were allowed to cool at room temperature and washed in tris-buffered saline containing 0.1% tween 20 (TBS-T). Next, slides were incubated with blocking buffer (Akoya Biosciences, NEL840001KT) for

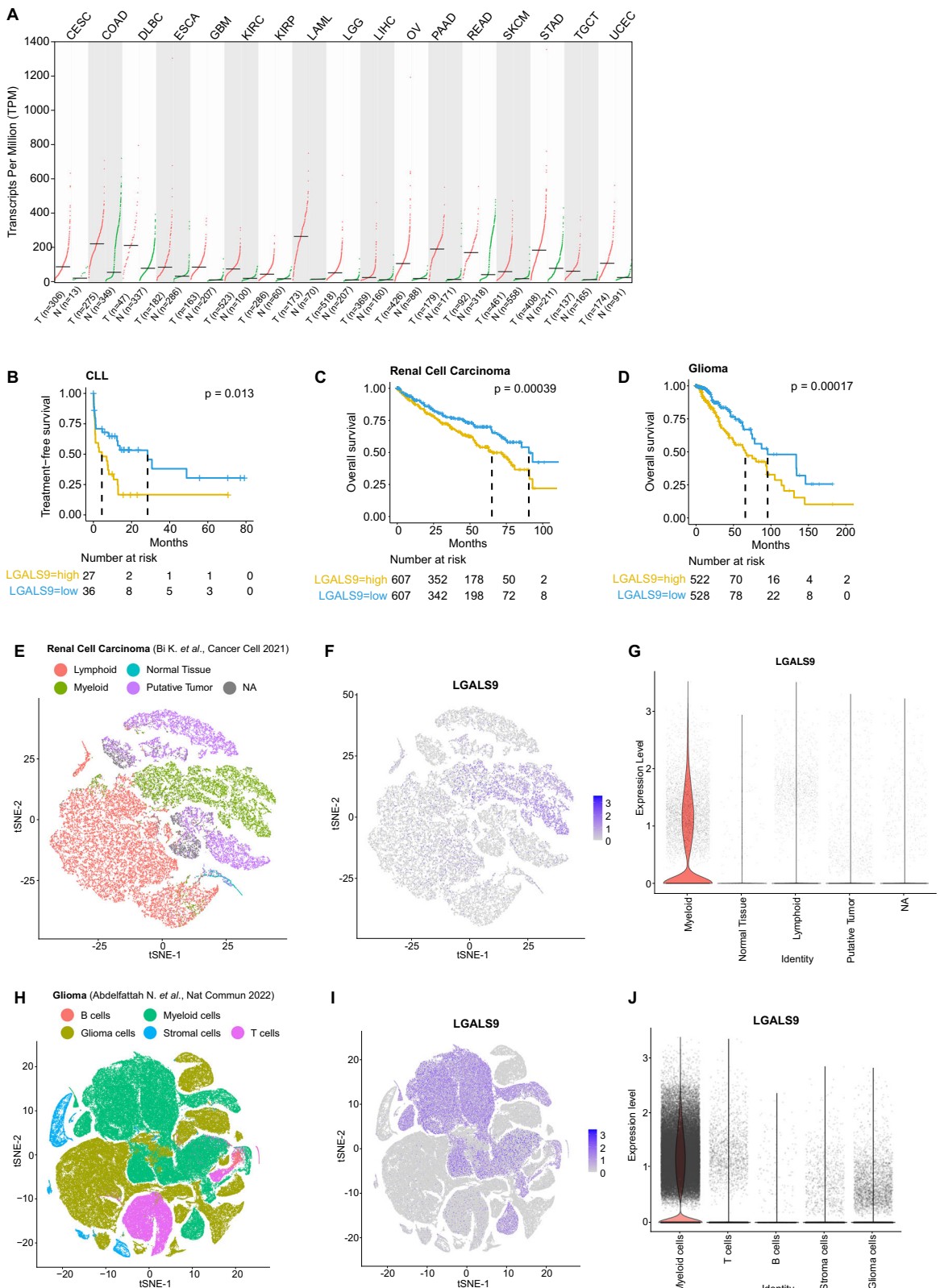

10 min at room temperature and incubated with the primary antibody for the indicated time and concentration (Supplementary Data 2). After three washing steps with TBS-T, slides were incubated with anti-rabbit HRP-conjugated secondary antibody (Akoya Biosciences, NEL840001KT). Sections underwent rounds of antigen stripping via incubation at 100 °C in AR9 buffer (Akoya Biosciences, AR900250ML) for 25 min, allowing removal of the excess of antibody complex

formed. For multi-color staining, tissue sections went through multiple rounds of blocking, primary and secondary antibody labelling, each round with a different Opal dye (Akoya Biosciences, NEL840001KT). Finally, background signal reduction through incubation for 10 min at room temperature in 0.1% Sudan Black (Sigma-Aldrich, 199664) took place and sections were washed thoroughly in TBS-T. Finally, nuclei were stained with DAPI (Akoya Biosciences, NEL840001KT) and slides

**Fig. 8 | Elevated galectin-9 expression correlates with poor survival in cancer patients. A** Differential expression of *LGALS9* in tumor versus healthy tissue in CESC (Cervical squamous cell carcinoma and endocervical adenocarcinoma), COAD (Colon adenocarcinoma), DLCL (Diffuse large B cell lymphoma), ESCA (Esophageal carcinoma), GBM (Glioblastoma multiforme), KIRC (Kidney renal clear cell carcinoma), KIRP (Kidney renal papillary cell carcinoma), LAML (Acute Myeloid Leukemia), LGG (Brain Lower Grade Glioma), LIHC (Liver hepatocellular carcinoma), OV (Ovarian serous cystadenocarcinoma), PAAD (Pancreatic adenocarcinoma), READ (Rectum adenocarcinoma), SKCM (Skin Cutaneous Melanoma), STAD (Stomach adenocarcinoma), TGCT (Testicular Germ Cell Tumors), and UCEC (Uterine Corpus Endometrial Carcinoma) analyzed by the standard processing pipeline GEPIA2 with default cut-off settings[49]. Statistical differences were assessed by limma model with adjusted *p*-values (Benjamini-Hochberg FDR). **B** Time-to-treatment in CLL patients with high or low galectin-9 protein levels (*n* = 63)[50]. Differences were assessed using Cox proportional hazard model. **C, D** Overall survival in renal cell carcinoma (**C**) and glioma (**D**) patients with high or low *LGALS9* transcript levels. Differences in survival were assessed using Cox proportional hazard model with Benjamini-Hochberg correction. **E–J** Single-cell RNA-seq analysis of tumor samples in renal cell carcinoma[104] (**E–G**) and glioma[105] (**H–J**). **E, H** UMAP plots identifying tumor, infiltrating immune cells and normal tissue cells from (**E**) 8 renal cell carcinoma patients, and (**H**) 9 glioma patients. **F, I** UMAP plot displaying *LGALS9* expression. **G, J** Violin plots of *LGALS9* expression in the different cell types. Source data are provided as a Source Data file Fig8.

were mounted using ProLong Diamond Antifade Mountant (Life Technologies, P36965).

Slides were imaged using a Carl Zeiss Axioscan 7 slide scanner at 20X.

### Immunofluorescence staining and imaging of CLL TMA
TMA 159/2 was de-paraffinized and subjected to antigen retrieval for 5 min at 95 °C followed by 5 min at 107 °C, using pH8.5 EZ-AR 2 Elegance buffer (BioGenex). To reduce tissue autofluorescence, slides were placed in a transparent reservoir containing 4.5% H$_2$O$_2$ and 24 mM NaOH in PBS and illuminated with white light for 60 min followed by 365 nm light for 30 min at room temperature[77]. Slides were rinsed with surfactant wash buffer (0.025% Triton X-100 in PBS), placed in a humidified stain tray, and incubated in Image-iT FX Signal Enhancer (Thermo Fisher) for 15 min at room temperature. After rinsing with surfactant wash buffer, the slides were placed in a humidity tray and stained with the panel of fluor- and hapten-labeled primary antibodies in PBS-Antibody Stabilizer (CANDOR Bioscience GmbH) containing 5% mouse serum and 5% rabbit serum for 2 h at room temperature. Slides were then rinsed again with surfactant wash buffer and placed in a humidified stain tray and incubated with SYTOX Blue (Thermo Fisher), ArgoFluor 845 mouse-anti-DIG in PBS-Antibody Stabilizer containing 10% goat serum for 30 min at room temperature. The slides were then rinsed a final time with surfactant wash buffer and PBS, coverslipped with ArgoFluor Mounting Media (RareCyte, Inc.) and dried overnight[78].

Slides were imaged using an Orion instrument (RareCyte, Inc.) at 20X. Raw image files were processed to correct for system aberrations; then signals from individual targets were isolated to separate channels using the Spectral Matrix obtained with control samples, followed by stitching of FOVs to generate a continuous open microscopy environment (OME) pyramid TIFF image.

### Image analysis
**Preprocessing.** TMA 159/2 with 18-color staining was analyzed using the multiple-choice microscopy pipeline MCMICRO[79] (Commit 91fcd5eefd69da112414b06cf3a65a9a66afeccf). Dearraying was performed by Coreograph[79] 2.2.8, Unmicst[80] 2.7.0 segmented the cells based on the nuclear channel Sytox, S3segmenter[81] 1.3.12 generated single-cell masks and Mcquant[81] 1.5.1 quantified the signal of each of the 18 channels in each cell. The corresponding parameters (params.yml) for full reproducibility of this MCMICRO runs are available on Github (https://github.com/SchapiroLabor/ImageAnalysisTMA159_2).

Following visual inspection of Coreograph's QC file, all cores that were damaged were filtered out. Additionally, the 4 muscle tissue cores – which were used as control – were filtered. In total, we performed the downstream analysis with 42 cores. Channel numbers 2 (Sytox), 14 (E-cadherin), 17 (Pan-CK) and 18 (AF-Tissue) were removed for downstream analysis since (i) Sytox and AF-Tissue represent nuclei and autofluorescence which are not relevant for cell type calling; and (ii) E-cadherin and Pan-CK expression are not expected in LN tissue and therefore, failed our visual QC due to unspecific staining patterns. Image visualization was performed with QuPath[82] and Napari[83].

**Batch correction.** Cell type clustering on the raw data results in clusters consisting mainly of cells of a single core of origin. This is due to preanalytical variability, which requires batch correction on core-level. Therefore, marker intensities were normalized with Mxnorm[84] 0.0.0.9000 (transform: log10_mean_divide, method: none) before using Rphenograph[85] 0.99.1 to cluster the individual cell types.

**Cell type assignment.** Cell types were assigned based on marker thresholding (positive/negative) and comparison to an expert-curated list of markers associated with the specific cell types (Supplementary Data 3).

**Spatial Analysis.** Neighborhood analysis was performed with Giotto 1.1.2[18]. After building a Delaunay network and running 1000 simulations, we compared the number of observed interactions between the cell types with their respective expected number of interactions. Cell-cell interactions between the same cell types are significantly enriched because of segmentation inaccuracies. Cells are slightly over segmented creating multiple cells with the same phenotype.

**Statistical analysis.** The subsequently calculated cell type frequencies in each core were used to perform two-sided t-tests (base package stats) for changes in cell type frequencies between patients that were positive or negative for a variety of binary clinical parameters (sex, *IGHV*, del13q, del17p, died, treated, del17p_tp53 (del17p and/or tp53 positive)). Additionally, Cox regression (survivalAnalysis 0.3.0) was calculated on overall survival and time to next treatment. Finally, correlation with the Pearson method was checked (cor.test() in base package stats) between overall survival, time to next treatment and all cell type frequencies.

Up to the statistical analysis, calculations were performed under R version 4.1.0 (2021-05-18) on platform x86_64-apple-darwin17.0 (64-bit) running under macOS 13.0. Statistical analysis was done on R version 4.0.5 (2021-03-31) on platform: x86_64-apple-darwin17.0 (64-bit) running under: macOS Big Sur 10.16.

All the mentioned steps of image processing were performed on the BWforCluster.

### Mass cytometry
A panel of 42 heavy metal-labeled antibodies for the detection of both surface and intracellular proteins was adopted from Bengsch et al. and modified based on literature research in order to characterize diverse T-cell phenotypes[86]. The complete list of proteins detected and the heavy metal-conjugated antibodies used are listed in the Supplementary Data 2. For most of the markers, heavy metal-conjugated antibodies were commercially available and purchased from Fluidigm. Where no heavy metal-conjugated antibodies were commercially

available, coupling of heavy metal to respective antibodies was performed in-house using the Maxpar X8 Multimetal Labeling kit (Fluidigm) following the manufacturer's instructions. ¹¹³In, ¹¹⁵In and ¹³⁹La heavy metal isotopes (#) were not available from Fluidigm and were purchased from Trace Sciences International (¹¹³In) and Sigma (¹¹⁵In, #203440-1 G; ¹³⁹La, #211605-100 G). Following the preparation of a 1 M solution, ¹¹³In, ¹¹⁵Ln and ¹³⁹La heavy-metal isotopes (#) were conjugated to monoclonal purified IgG antibodies using the Maxpar X8 Multimetal Labeling kit (Fluidigm, #201300).

Frozen cell suspensions were thawed and washed prior to incubation in 15 mL of pre-heated RPMI 1640 containing 10% FBS in a roller incubator for 30 min at room temperature. Cells were filtered to remove dead cells and B-cell depletion was performed using human CD19 microbead (Miltenyi Biotec) labeling according to the manufacturer's protocol. Unlabeled CD19⁻ cells were collected, washed, filtered, and counted prior to mass cytometry processing.

CD19-depleted single-cell suspensions were stained for mass cytometry analysis[37]. Briefly, cells were stained with 5 µM cisplatin (Cell-ID Cisplatin, Fluidigm) for 5 min in order to label dead cells. Cells were then washed once and cell surface staining was performed for 30 min at room temperature. After a washing step, cells were fixed with Fixation/Permeabilization buffer (Thermo Fisher Scientific) following the manufacturer's instructions. Intracellular staining was performed by incubating the antibody-cocktail for 30 min at room temperature. After a washing step, cells were stained with cell-ID Intercalator-Ir (Fluidigm) in fixation and permeabilization solution, followed by another two washing steps with PBS and ddH₂O, respectively. Prior to acquisition, cells were resuspended at a concentration of $5 \times 10^5$ cells/mL in ddH₂O with 1:10 calibration beads (EQ Four Element Calibration Beads, Fluidigm). Samples were analyzed at a flow rate of 0.03 mL/min with the Helios mass cytometer (Fluidigm) of the National Cytometry Platform (LIH). Initial data processing and quality control were performed. Flow cytometry standard (FCS) files were normalized with EQ beads using the HELIOS instrument acquisition software (Fluidigm).

As a first analysis step, samples were pre-gated as follows: gates were placed on cells (Beads vs. Ir191), singlets (Ir193 vs. Ir191) and live cells (Pt195⁻). Next, non-B immune cells (CD19⁻ CD45⁺) and CD3⁺ T cells (CD3⁺ CD56⁻) were selected and exported as fcs files.

Raw fcs data files were merged into a flowSet object using flowCore package[87]. Signal intensities for each marker were arcsinh-transformed using a co-factor of 5 (default)[88]. Sample quality was assessed based on cell counts per sample and samples with less than 1500 cells were excluded. Multi-dimension scaling (MDS) plotting using median arcsinh-transformed marker expression for all cells in each sample was used for sample clustering analysis.

Cytometry data analysis tools (CATALYST) R package[89] containing FlowSOM[90] and ConsensusClusterPlus[91] metaclustering methods was used for cell cluster identification using all cells from all samples. Clustering was performed using arcsinh-transformed expression of 33 markers, i.e., excluding the markers DNA content and cisplatin viability, which were used to select viable, single cells, and CD19 and HLA-DR, which were used to exclude B cells (Supplementary Data 2). TIM3, LAG3, GITR, CD47 and 4-1BB were excluded from the analysis due to low signal and likely adding noise in the cluster generation process.

The maximum number of clusters allowed to be evaluated was set to maxK = 30, after biological relevance of obtained clusters was assessed, and k < 30 and k > 30 were verified to be underfitting or overfitting, respectively. Cell clustering was visualized using the UMAP algorithm, displaying $1 \times 10^3$ random cells from each sample. The robustness of this approach was evaluated by repeatedly plotting an increasing number of randomly picked cells from each sample, with a range of 200 to 1000 cells, and evaluating the similarity of the 30 subpopulations recognized in the respective UMAP plots. Sample #HD3 was excluded as an outlier.

## Cell sorting for single-cell RNA and TCR sequencing

Single-cell suspensions from PB and LNs of CLL patients were retrieved by partially thawing vials of cryopreserved cells in order to preserve unused cells. Samples were stained for cell surface proteins for 30 min. After washing with PBS, cells were resuspended in PBS + 5% FBS containing 0.2 µg/mL DAPI prior cell sorting. The gating strategy for CD3⁺ T-cell and CLL-cell sorting is depicted in Supplementary Fig. 5A. Cells were sorted in PBS + 2% FBS using a BD FACSAria II or BD FAC-SArisa Fusion (both from BD Biosciences) cell sorters. The purity of cells after sorting was above 95%.

## Single-cell RNA sequencing library construction using the 10x Genomics Chromium platform

Five CLL LN and five PB samples were processed for single-cell TCR and 5′ gene expression profiling of CD3⁺ T cells and CLL cells using the Chromium Next GEM Single Cell V(D)J Solution from 10X Genomics following the manufacturer's instructions. Briefly, for a target cell recovery of 5000 cells, the concentration of T cells was adjusted to $1 \times 10^3$ cells/µL and tumor cells were added at a ratio of 1:20. Cells were then loaded into the Single Cell A Chip (10X Genomics). GEX libraries were sequenced on a HiSeq 4000 machine (Illumina) or on a NovaSeq 6000 (Illumina), and V(D)J-enriched libraries were sequenced on a NextSeq 550 (Illumina).

## Single-cell RNA-Seq data processing

Sequencing reads were aligned by Cell Ranger (5.0.1) to reference version hg38(2020-A) for human data and to reference version mm10(2020-A) for mouse data[92]. Raw count matrices were processed by the quality control pipeline to determine the optimal cut-offs for filtering. Cells were filtered by read counts and mitochondrial RNA content. Lower cut-off for read counts were determined by read-counts distribution for samples in each dataset and were set at 350 for human and 500 for mouse, mitochondrial RNA content should be lower than 10%.

Raw gene count matrices were processed by Seurat(v4.1.0)[21], and expression values were normalized and scaled. Doublets were filtered by DoubletFinder(2.0.3)[93]. Dimensional reduction was performed by Principle Component Analysis (PCA). Datasets were integrated by HARMONY[94] to reduce batch bias on calculated principal components. Sample identity was used as the batch variable for HARMONY integration. Uniform Manifold Approximation and Projection (UMAP) was used for dimensional reduction on the first 23 HARMONY computed cell-embeddings.

Cluster were defined by the Louvain method at the resolution of 0.8, clusters defined at this step were used for annotation by manual curation. Markers for each cluster were produced by differential expression analysis searching for overexpressing markers of each cluster.

## Pseudotime analysis

Diffusion pseudotime analysis of LN T cells was performed using destiny (3.8.1)[25]. Cells expressing CD4 and CD8 were analysed separately. To retrieve the pseudotime of each marker differentiated cell groups, DPT function from destiny was used for computing the diffusion pseudotime on each cell. For each cell type, the root nodes were placed within the corresponding naïve cell type cluster.

To performed differential gene expression analysis on pseudotime, the diffusion pseudotime values of cells in trajectories were analysed together with their transcript expression values by tradeSeq[95]. Differentially expressed genes were plotted against ranked diffusion pseudotime for illustration.

## TCR data analysis

V(D)J transcripts from single cells were aligned and counted using the Cell Ranger pipeline (5.0.1). GRCm38 VDJ Reference 5.0.0 from 10X

Genomics was used for mouse data and GRCh38 VDJ Reference 5.0.0 from 10X Genomics was used for human data. In addition, an output file containing TCR α- and β-chain CDR3 nucleotide sequences and a cell barcode for all single cells was generated. Only productive rearrangements were evaluated, and 2 or more cells containing the same α- and β-chain CDR3 consensus nucleotide sequences were considered cell clones. scRepertoire[96] was used to process contig and clonotype information from Cell Ranger. The summarized TCR information per cell was merged with Seurat object using the shared cell barcodes between the TCR library and the RNA-Seq library for integrated analysis. The Shannon diversity index was computed to represent the diversity of the TCR repertoire in each single-cell cluster. Cells with expanded TCRs were categorized by their expansion level and then illustrated by UMAP plots.

### Prediction of tumor-reactive TCR clonotypes

Our recently developed pipeline predicTCR[27] was applied to annotate tumor-reactive TCR clonotypes using Seurat. Briefly, the gene expression was normalized using SCTransform. Normalized data was then imported into Python and reactivity was predicted using predicTCR model. The probability of reactivity was then averaged for each TCR clonotype (as predicTCR score) and the threshold was determined using Jenk-Fisher natural break optimization. Clonotype with an average probability of reactivity higher than the threshold was designated as tumor-reactive and vice versa.

### In silico HLA typing from RNA-seq data

The arcasHLA tool[97] was used to perform in silico human leukocyte antigen (HLA) typing using the single-cell sequencing fastq files as input.

### Immune repertoire sequencing

The TCR repertoire for T cells from the lymph nodes was generated using the DriverMap Adaptive Immune Receptor (AIR) TCR-BCR profiling kit from Cellecta, following the manufacturer's instructions. DNA libraries could be constructed for material from patients BC0, BC1, BC3, and BC9. Libraries were sequenced on an Oxford Nanopore Technologies PromethION Flow Cell. TCR sequences were constructed using miXCR v4.6.0[98].

### Immune repertoire analysis

TCR repertoires were screened for known reactivities using exact CDR3 matches using VDJdb[99], and Immunewatch detect v1.0 (ImmuneWatch DETECT, Version 1.0. Developed by ImmuneWatch BV. 2024. Available at: https://www.immunewatch.com/detect), using a minimum score of 0.2 as in the developer's instructions.

Convergent groups of TCRs were calculated using the GLIPH2 algorithm[28] as implemented in R v4.4.1 (R Core Team, 2022) using the turboGliph package (https://github.com/HetzDra/turboGliph) using the default settings. Data were collated in R using custom scripts and visualised using Cytoscape v3.10.3[100].

### Ligand–receptor interaction analysis

CellChat[32] was used to infer and visualize the cell-cell communication network in CLL LNs. To find differential ligand–receptor interactions between CLL LNs and control LNs, a control dataset derived from 5 reactive lymph nodes[34] was added for comparison. The reactive lymph node dataset was independently processed by the same quality control pipeline and single-cell analysis pipeline. CellChat was used for differential cell-cell communication analysis. Ligand–receptor interactions from the OmnipathR database[101] were included as candidates. CellChat analysis was first performed independently on the two datasets using LIANA[102]. The results were then contrasted by CellChat for differential interactions. Labels from the two comparing datasets were

harmonized to 8 major cell types (CLL/B, CD4 $T_{CM}$, $T_{REG}$, CD4 $T_{EM}$, CD8 $T_{EM}$, CD4 $T_N$, CD8 $T_N$, $T_{FH}$) for comparison. Differential Interactions with p-values below 0.01 were visualized by the dot plot functions of CellChat.

### Expression of galectin-9 protein and its impact on survival in CLL

Galectin-9 protein levels in leukemic cells from the blood of CLL patients were obtained from Herbst et al.[50]. Mean relative galectin-9 levels were compared between IGHV-mutated and -unmutated CLL patients using a two-sided Wilcoxon signed-rank test. Time to next treatment (TTNT) was calculated from the date of sample collection to subsequent treatment initiation. Proportional hazards regression (Cox regression) was used to calculate the impact of galectin-9 abundance on TTNT using the R package survival (version 3.2-3). Kaplan–Meier curves were plotted for all samples and for IGHV-mutated and -unmutated CLL patients separately.

### Expression of LGALS9 and its impact on survival in TCGA samples

Expression of LGALS9 in TCGA samples was gathered with R (4.3.1) and Bioconductor using library RTCGA.rnaseq (1.3.0,2015-11-01) and RTCGA.clinical (1.3.0,2015-11-01). Survival curves were prepared by survminer (0.4.9), splitting the samples based on the median LGALS9 expression into two groups. Differences in survival were assessed using the Cox proportional hazard model and multiple testing was corrected using Benjamini-Hochberg FDR. Using a standard processing pipeline and the default cut-off GEPIA2 was used to compare tumor and normal samples from TCGA and the GTEx projects[103].

### Expression of LGALS9 in single-cell RNA-seq data sets

The expression of LGALS9 was assessed in renal cell carcinoma (SCP1288)[104] and glioblastoma (SCP1985)[105]. Data was collected from (https://singlecell.broadinstitute.org/single_cell) and processed in R using Seurat (4.9.9.9044). In short, data was loaded using Read10X and after the creation of a Seurat object (min.cells=3, min.features=200) processed using default values with NormalizeData, FindVariableFeatures, ScaleData, RunPCA, FindNeighbours, FindClusters, and RunTSNE. Figures were prepared using FeaturePlot and DimPlot, highlighting cell types. Violin plots were prepared using ggplot2 (3.4.2).

### Reporting summary

Further information on research design is available in the Nature Portfolio Reporting Summary linked to this article.

## Data availability

The human single-cell RNA- and TCR-sequencing data including matrix files generated in this study have been deposited in the European Genome-phenome Archive (EGA) under the accession ID EGAS00001006864. The sequencing raw data are available under restricted access, which can be obtained by request to the Data Access Committee. Mouse single-cell RNA- and TCR-sequencing data are available on Gene Expression Omnibus under the accession ID GSE221395. The CyTOF data used in this study are available in the Zenodo database under the https://doi.org/10.5281/zenodo.15606759. All other data are available in the article and its Supplementary files or from the corresponding author upon request. Source data are provided with this paper.

## Code availability

All code related to CyTOF and scRNA-seq analyses is available on GitHub (https://github.com/mzapatka/Tcell_CLL).

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

## Acknowledgements

The authors would like to thank all members of the High Throughput Sequencing Unit of the Genomics and Proteomics Core Facility, of the Light Microscopy Core Facility, and of the Single-cell Open Lab (scOpenLab) at the German Cancer Research Center (DKFZ, Heidelberg) for their technical assistance. We would like to thank the Immune Monitoring Unit (IMU) at the NCT Heidelberg for preparing the Adaptive Immune Receptor sequencing libraries. We also thank the National Cytometry Platform (NCP) of the Luxembourg Institute of Health (LIH, Luxembourg) for assistance with the generation of cytometry data, in particular Maira Konstantinou and Dominique Revets. The NCP is supported by funding from Luxembourg's Ministry of Higher Education and Research (MESR). Tissue microarrays and lymph node sections were provided by the Tissue Bank of the National Center for Tumor Diseases (NCT) Heidelberg, Germany in accordance with the regulations of the tissue bank and the approval of the ethics committee of Heidelberg University.

This study was supported by a research grant of the German Cancer Aid (Deutsche Krebshilfe, grant 70114114) and the Euranet Transcan-3 project ImmuMet to M.S., by grants from FNRS-Télévie to I.F.B. (7.4529.19, 7.6603.21), M.W. (7.6504.18), G.P. (7.4501.18, 7.6518.20), S.G. (7.4502.19, 7.6604.21), and from the Luxembourg National Research Fund (FNR), Fondation Cancer and Plooschter Projet to E.M. and J.P. (C20/BM/14582635, C20/BM/14592342, and C23/BM/17987391). E.C. was supported by the "la Caixa" Foundation Health Research 2022 Program (CLLSYSTEMS LCF/PR/HR22/52420015 (HR22-00172). C.L.T. was supported by a DKFZ Hector Seed Funding Kick-start EarlyCareer fellowship (C-3PO). The authors gratefully acknowledge the data storage service SDS@hd supported by the Ministry of Science, Research and the Arts Baden-Württemberg (MWK) and the German Research Foundation (DFG) through grant INST 35/1503-1 FUGG. The authors acknowledge support by the state of Baden-Württemberg through bwHPC and the German Research Foundation (DFG) through grant INST 35/1597-1 FUGG. M.C., K.B., and D.S. were supported by the German Federal Ministry of Education and Research (BMBF 01ZZ2004). D.S. was additionally supported by the Ministry for Science, Research and Science Baden-Württemberg "AI Health Innovation Cluster" and "MULTI-SPACE"; research funding from Cellzome, a GSK company, the Bruno and Helene Jöster Stiftung, and the Carl-Zeiss-Stiftung through the Multi-dimensionAI project (CZS-Project number: P2022-08-101).

## Author contributions

L.L.-C. designed the study, performed experiments, analyzed data, performed bioinformatical analyses, generated figures, and wrote the paper. J.K.L.W. performed bioinformatical analyses, generated figures, and wrote the paper. I.F.B. and M.W. conceptualized part of the study, performed experiments, and analyzed data. Y.P. analyzed scRNA-seq data and CyTOF data. L.-M.P. generated scRNA-seq libraries. A.F. conducted experiments, analyzed data, and generated figures. S.G., G.P., C.S. A.C., F.C., P.-M.B., and D.E.C. performed experiments and analyzed data. M.C., K.B. and D.S. analyzed multiplex imaging data, and generated figures. M.I. performed bioinformatical analyses and generated figures. C.L.T. and E.W.G. performed predicTCR and immune repertoire analyses and generated figures. T.R. and S.D. provided input regarding cluster annotations and performed correlation analyses. J.P.M. established the pipeline for single-cell RNA-sequencing. E.G.-H. and A.R. supported the study with lymph node sections and expertise in interpreting imaging data of lymph nodes. D.C., E.C., and S.D. provided clinical samples and information. P.L. provided logistic and budget support and was involved in scientific discussions. E.M., J.P., M.Z., and M.S. conceptualized the study, guided experiments and data analysis, provided logistic and budget support, and wrote the paper.

## Funding

## Competing interests

D.S. reports funding from GSK and received fees/honoraria from Immunai, Noetik, Alpenglow and Lunaphore. K.B. reports fees from Lunaphore. C.L.T. and E.W.G. hold patents and PCT applications describing methods to identify tumor-reactive T cells, and are founders or employees of Tcelltech GmbH. All other authors declare no competing interests.

## Additional information

[1]Division of Molecular Genetics, German Cancer Research Center (DKFZ), Heidelberg, Germany. [2]Tumor Stroma Interactions, Department of Cancer Research, Luxembourg Institute of Health, Luxembourg, Luxembourg. [3]Faculty of Biosciences, University of Heidelberg, Heidelberg, Germany. [4]CCU Neuroimmunology and Brain Tumor Immunology, German Cancer Research Center, Heidelberg, Germany. [5]Institute for Computational Biomedicine, Faculty of Medicine, Heidelberg University and Heidelberg University Hospital, Heidelberg, Germany. [6]Department of Internal Medicine V, Hematology, Oncology and Rheumatology, University Hospital Heidelberg, Heidelberg, Germany. [7]European Molecular Biology Laboratory (EMBL), Heidelberg, Germany. [8]Molecular Medicine Partnership Unit (MMPU), Heidelberg, Germany. [9]scOpenLab, German Cancer Research Center, Heidelberg, Germany. [10]National Cytometry Platform, Luxembourg Institute of Health, Luxembourg, Luxembourg. [11]RareCyte, Seattle, WA, USA. [12]Institute for Pathology, University of Würzburg, Würzburg, Germany. [13]Institut d'Investigacions Biomèdiques August Pi i Sunyer (IDIBAPS), Hematopathology Unit, Hospital Clinic, CIBERONC, Barcelona, Spain. [14]Translational Spatial Profiling Center (TSPC), Heidelberg University Hospital, Heidelberg, Germany. [15]Institute of Pathology, Faculty of Medicine, Heidelberg University and Heidelberg University Hospital, Heidelberg, Germany. [16]These authors contributed equally: L. Llaó-Cid, JKL Wong, E. Moussay, J. Paggetti, M. Zapatka, M. Seiffert ✉e-mail: m.seiffert@dkfz.de

