## [Transparent Peer Review file · Nature Communications]

Integrative Multi-Omics Reveals a Regulatory and Exhausted T-Cell Landscape in CLL and Identifies Galectin-9 as an Immunotherapy Target

Corresponding Author: Dr Martina Seiffert

Parts of this Peer Review File have been redacted as indicated at the request of the authors.

A version of this paper was originally rejected for publication by Nature Communications, however that decision was reconsidered after appeal by the authors.

Version 1:

Reviewer comments:

Reviewer #1

(Remarks to the Author)

The manuscript provides a thorough overview of the T cell landscape of chronic lymphocytic leukemia, achieved via multi-omic integration single-cell RNAseq datasets, mass cytometry and tissue microarrays from CLL lymph nodes, peripheral blood and bone marrow samples. This work represents a unique and novel resource for the CLL field and offers many new insights into T cell subset composition and dysfunction. The Galectin 9 discovery and in vivo studies provide important implication for future studies aimed at improving immunotherapeutic outcomes in this disease setting. Methodology is sound and methods are sufficiently detailed to allow reproduction of the work.

Reviewer #2

(Remarks to the Author)

General comments:

The manuscript investigates the T cell landscape in chronic lymphocytic leukemia (CLL) using samples from patients and a CLL mouse model. Through single-cell RNA sequencing, mass cytometry (CyTOF), and tissue microarray analysis, the study delineates T cell phenotypes and transcriptional programs across different tissues, where main samples are from lymph nodes, with additional comparisons to peripheral blood and bone marrow.

The main applaud of the manuscript is the use of lymph node as a sampling material, and to me, the number of CLL and healthy LN is impressive. All in all, the manuscript is cleraly written, the methods are sound and the conclusion drawn from the data are well based. However, I have some points that the authors could consider to address knowledge gaps in the field.

Major:

1. As stated in the Introduction, patients with CLL do not benefit from current ICIs. Based on their data, how are the authors explaining this? There are not enough exhausted cells, the exhausted cells are too terminally exhausted to be rescued, too many suppressing elements like Tregs, or there are not enough target antigens for the T cells?

Based on Fig 3C, the LN is the site of main exhaustion, but even in there, based on Fig 3D it seems that the median % of CD8 TEX CD39 (based on CD39, most likely the tumor reactive T cells?) is merely 1 %, and the CD8 TEX CD38 3 %? I would also want to see a clear representation of the expression of the main exhaustion markers, PD1, LAG3, TIGIT, TIM3, CTLA4 in PB, BM and LN of CLL and healthy.

2. To me, the reason why anti-PD1 does not work in CLL might arise from studying Richter's transformation, where anti-PD1 is efficacious in some patients. Do the authors have access to RT patients LN or some other tissue, or are there perhaps already published data?

3. Clinical association and LN immune cells. With such a small sample size (n=20), one perhaps would not expect to find meaningful association after the adjusted P values. Although I applaud the authors for the correct way of doing the adjusting, were there any findings that were significant before the comparisons? Especially interesting would be to see some effect size, e.g., log2 fold-change between the clinical categories.

4. TP53/del 17p is clinically one of the most interesting populations. I see the authors have only three such samples, BC1, BC12 (and BC15?), but for many samples the analysis is not available. Could the authors retrieve TP53/del17p status for all the samples? While clinically a physician might not study the TP53 study if they are not planning a treatment, at least the treated patients should have this information available. There are many reasons to believe that patients with these abnormalities might have a distinct immune system that is more malleable with immunotherapies.

5. Comparing T cells phenotypes from LN to PB with TCRseq. Majority of CLL studies, have focused on circulating CLL and non-CLL immune cells. As the authors have from some individuals both PB and LN, it would be crucial to deliver what are the disease biomarkers that are only seen in LN, what are seen in PB, and what are seen in both. To me, this should be done with scRNA+TCRab-seq. Could the authors do additional scRNA+TCRab-seq to follow T cells clones from lymph node in peripheral blood? Of course following these T cells in bone marrow would be optimal but the process might be too invasive. Especially the BC1 with TP53mut and hyperexpanded clone would be informative.

6. T cell antigens. Do the authors have any hints what are their most interesting T cells targeting? Tumor associated antigens, neoantigens (somatic mutations / CNVs); what are known, what are the authors thoughts? Please discuss. Could the authors do some unsupervised clustering of the TCRs within / across individuals to have an intuition how many antigens the TCR might be targeting? Some algorithms could be e.g., GLIPH2, TCRdist.

7. CLL LN vs healthy LN scRNA-seq. It is known that CD4+ T cells do not expand as much as CD8+ T cells, so I would say that 4-26 % of clonal expansion of CD4+ T cells are major, unlike the authors. Could the authors compare their CLL LN scRNA+TCRab-seq data to healthy LN scRNA-seq data, like they have in the interaction analyses? This could put the phenotypes into perspective, and hopefully the clonal expansion rates. Ideally these samples should be somewhat age-matched, as that is the major determinant.

8. LGALS9 instead of TIM3. Why would the authors target LGALS9 instead of TIM3, which is an immune checkpoint and is/was of great interest for hematologists in MDS/AML with sabatolimab? To me, the data analysis on solid cancers is obscure and the OS data with 50 CLL samples is not convincing that LGALS9 is a better target.

9. What are the main points learned from the IHC? To me, it seems that these data are underused.

Minor:

1. Discussion / intro related to CLL therapies. The most common CLL therapies; Obi-Ven, BTK, FCR; have distinct effects on the immune system - could the authors discuss how their findings are attributed to these?

2. Discussion / intro related to ICI. I think the field has moved on from ICIs towards bispecifics and CAR T:s, with and without combinations (e.g., BTKs). I think the intro / discussion should reflect more of that, although I do understand that readers of Nat Comm might be more accustomed to ICIs.

3. LNs from same individuals. Could you comment on the LNs from the same donor? These are best biological replicates, as the host genetics and disease covariates are identical.

4. What is the main new information generated with the pseudotime analyses? What are the information what could've not been received otherwise, are there some therapeutic applications, is the data valuable with some in vitro / in vivo assays?

5. Cell types enriched in rLN. It would be helpful to at least in the supplements see what are the cell types that are enriched in rLN compared to CLL LN. For example, are TFH enriched to rLN?

6. TIM3 in CyTOF. I see that in the methods that the signal for TIM3 was too low to be acceptable for downstream analysis. Does that mean that it is not expressed enough or that the antibody does not work?

7. Lines 68 - 69: I find these lines confusing. I think the authors mean SLL and not CLL. Please revise.

8. Could NCAM be referred as CD56? I think people are more accustomed to that naming.

9. Line 430 Galectin-9

10. Figure 1A and C require greater resolution

11. Figure 3B would benefit from adding hovered broad labels for the islands, such as CD8+, CD4+ etc, know skipping

between Fig 1 and 3 is bit cumbersome.

Reviewer #3

(Remarks to the Author)

In this paper by Liào Cid and colleagues, the Authors investigated, by complementary approaches, the T cells landscape in the CLL microenvironment. The characterization of these subsets of cells allowed to identify a disease-specific accumulation of regulatory and exhausted populations. Moreover, by looking at the interactions between CLL and T cells, the Authors reported the identification of Galactin-9/TIM3 axis as relevant in driving and shaping CLL immune-suppression and immune-escape.

The paper is well written and organized. The reported findings are novel and provide significant insight in the understanding of CLL pathogenesis and responses to therapies, opening for novel targeting opportunities. However, there are several points that need to be addressed by the Authors.

Major points.

1. How do the Authors explain the low levels of TIM3 transcripts and its expression as a receptor expressed in exhausted T-cells? Are there any post-transcriptional mechanisms involved in their expression?
2. Assuming that there are differences in the ratio between regulatory and inflammatory blood cells between males and female specimens - as shown in the literature – is there a potential bias in using only female mice? This point has to be discussed.
3. In Figure 1B, in the UMAP plot, in the bottom left corner (-5 and -5 as coordinates) there is a little cluster of cells with the same color of cluster 1. Is it just noise or has the first cluster a subpopulation which is separated from the rest? If so, please comment and explain why they are distinct.
4. In Figure 1D, it seems that there is a pattern of exhaustion in T-reg cells instead of cytotoxic/effector T-cells. Referring to the statement reported at line 145-150, there seems to be a strong correlation between CD4+ T-reg and PD1+ CD8+ T-cells, which is in line with a role of CD4+ T-regs in promoting the accumulation of dysfunctional CD8+ T-cells. Do the Authors suspect the existence of a crosstalk between regulatory and cytotoxic cells as a primary cause of the exhausted phenotype? Indeed, the persistent antigenic stimulation, as well as a general transcriptional reprogramming, is regarded to be one of the main causes for T-cell exhaustion.
5. Referring to Figure 2B, although a really low p-value is calculated, I would not state that there is a strong correlation since the R2 is 0.37. The correlation is there, but not that strong. I would rephrase accordingly.
6. The Authors focused on CD4 PD1+-CD4 Tregs and CD8 PD1+-CD4 Tregs interaction, however, as shown in Figure 2D, there are other cells involved in heterotypic cell-interaction with greater/similar enrichment (e.g. CD4 PD1--CD4 Tregs). Why did the Authors focus only on the first one? Please provide an explanation.
7. In Figure 4D-E: are the statistical differences observed also statistically assessed? In other words, did the Authors also conduct a statistical analysis or is the significance already computed by the scoring methods?
8. Referring to Figure 5A, as only 5 patients were evaluated for the remaining part of the study, did the Authors use any tool to assess the statistical power of this setting? Were they the only patients at disposal? Was the choice of patients randomized? Did the Authors assess the presence of any confounding factors for the study? For instance, are there any genetic variants likely implicated in the crosstalk with the TME?
9. Referring to Figure 5B: what is the reason behind the NA values? Was the identification of the markers of clonal expansion not definable?
10. Referring to Figure 7B: how do the Authors explain the fact that CD4 Tn cells present similar level of expression markers to CD8 T cells?
11. In Figure 8A data are reported as TPM, but it would be better to plot them as TMM values. Indeed, TPMs are best suited for intra-sample comparisons. Alternatively, could the Authors provide a statistical test to assess the differences between healthy and patients?
12. Data included in Figure 6B are redundant with data plotted in Figure 6A: please remove them.
13. In the last section of the results, the Authors look at LGALS9 expression in other tumors than CLL. This part is somehow out of the topic of the paper, that is centered on CLL, and it would be better to keep it as a point of discussion, instead of results.
14. Given the interesting finding of galactin-9, it would be great to include this finding in the title of the paper.

Minor points, mainly related to graphical issues.

1. Figures 1A, C and 7E are blurred.
2. Figure 4G has “z-scores” cut in half.
3. Figure 2C has no piece of information about the blue staining (likely a DAPI for the nucleus)

Version 2:

Reviewer comments:

Reviewer #2

(Remarks to the Author)

I thank the authors for the responses to my comments. The following emerging issues need to be addressed:

1. Please tone down the claims of identifying CLL-reactive T cells based on analyzing gene expression profiles. The claims in their current forms are truly misleading, and the authors of predicTCR also recommend using their tool as a mere prioritization strategy for further antigen-reactivity screening. The analysis presented here is highly circular – predicTCR looks for gene expression of exhaustion related genes, so it is not surprising that the authors find the most “tumor reactivity” in cells expressing exhaustion related genes and that they are predominantly found in LN compared to PB. One of the main differences is also limited that predicTCR is trained on TILs, and I would argue that T cells in CLL LN are not the same as TILs, as the authors also state in their response to me.

The analysis in the current form is not in line what I asked, where I enquired about similarity within TCRs. The lines 334-351; are more in line what I ask, but would not highlight these in the abstract and would remove wild claims related to predicTCR. Especially lines 335-338 and 346-347 need to be removed, as I cannot understand how the authors claim validation for such a method without doing antigen-reactivity.

To be clear, I would remove the analyses performed with predicTCR as circular. Or then validate some of their hits with true antigen screening (e.g., by cloning TCRs and testing are they indeed reactive against the autologous CLL cells), but as hinted by the authors in the Discussion that is not in the immediate plans.

2. I believe data should be available in a non-raw format (e.g., gene-cell matrices) and additionally code should be also available.

Reviewer #3

(Remarks to the Author)

The Authors have addressed all the issues raised and modified the paper accordingly.
No more points from my side.

Version 3:

Reviewer comments:

Reviewer #2

(Remarks to the Author)

I thank the authors for their revised manuscript. They have addressed all my concerns.

REVIEWER COMMENTS

Reviewer #1 (Remarks to the Author):

The manuscript provides a thorough overview of the T cell landscape of chronic lymphocytic leukemia, achieved via multi-omic integration single-cell RNAseq datasets, mass cytometry and tissue microarrays from CLL lymph nodes, peripheral blood and bone marrow samples. This work represents a unique and novel resource for the CLL field and offers many new insights into T cell subset composition and dysfunction. The Galectin 9 discovery and in vivo studies provide important implication for future studies aimed at improving immunotherapeutic outcomes in this disease setting. Methodology is sound and methods are sufficiently detailed to allow reproduction of the work.

Response: Thank you very much for the thorough review of our manuscript and this positive comment. We are happy that you appreciate our novel data and their potential translational implications.

Reviewer #2 (Remarks to the Author):

General comments:

The manuscript investigates the T cell landscape in chronic lymphocytic leukemia (CLL) using samples from patients and a CLL mouse model. Through single-cell RNA sequencing, mass cytometry (CyTOF), and tissue microarray analysis, the study delineates T cell phenotypes and transcriptional programs across different tissues, where main samples are from lymph nodes, with additional comparisons to peripheral blood and bone marrow.

The main applaud of the manuscript is the use of lymph node as a sampling material, and to me, the number of CLL and healthy LN is impressive. All in all, the manuscript is cleraly written, the methods are sound and the conclusion drawn from the data are well based. However, I have some points that the authors could consider to address knowledge gaps in the field.

Response: Thank you very much for the thorough review of our manuscript and this positive comment. We are happy that you appreciate our sample collection and the data we produced of it.

Major:

1. As stated in the Introduction, patients with CLL do not benefit from current ICIs. Based on their data, how are the authors explaining this? There are not enough exhausted cells, the exhausted cells are too terminally exhausted to be rescued, too many suppressing elements like Tregs, or there are not enough target antigens for the T cells?

Response: Thank you for your insightful comment. Based on the data we present in our manuscript, we conclude that the lack of clinical benefit from ICIs in CLL patients is likely due to a combination of multiple factors. First, while there are exhausted T cells in CLL, a sign that there is active immune

response ongoing, a fraction of these cells is terminally exhausted and thus less amenable to rescue by ICIs. Additionally, the tumor microenvironment in CLL is enriched with immunosuppressive elements, such as Tregs, which further dampen the immune response. Finally, CLL cells may present fewer or less immunogenic target antigens for T cells to recognize. These factors, working together, likely contribute to the limited efficacy of ICIs in this setting. We hope this clarifies our perspective, and we made sure to elaborate on these points in the revised manuscript.

Updated Introduction on page 4: A lack of fit and functional T cells is discussed as a main limitation explaining the failure of immunotherapy responses in CLL and beyond^{4,5}. Chronic exposure of T cells to tumor (neo)antigens and an immune suppressive microenvironment leads to their exhaustion and dysfunction⁶. In addition, current treatment regimens for CLL were shown to impact T-cell fitness⁷.

Based on Fig 3C, the LN is the site of main exhaustion, but even in there, based on Fig 3D it seems that the median % of CD8 TEX CD39 (based on CD39, most likely the tumor reactive T cells?) is merely 1 %, and the CD8 TEX CD38 3 %? I would also want to see a clear representation of the expression of the main exhaustion markers, PD1, LAG3, TIGIT, TIM3, CTLA4 in PB, BM and LN of CLL and healthy.

Response: In comparison to solid cancers, where the main infiltrating T cells are likely tumor-reactive, the lymph node is a tissue highly abundant in naïve CD4 and CD8 T cells and follicular helper cells. Therefore, the frequency of CLL-reactive T cells is expected to be lower compared to cancers in other tissues. In addition, we observed a high degree of inter-patient variability in the abundances of the different T-cell clusters in the lymph nodes (shown in Fig. S4A) which can be partly explained by the variation in age (ranging from 46 to 77 years) and in tumor load in the tissue (ranging from 33 to 92 %). This information is now summarized in Suppl. Table 1. As suggested, we have now included an additional Figure 3E showing the expression of the activation/exhaustion markers PD1, TIGIT, CD39, CTLA4, OX40, EOMES and TOX in CD8 T_{EM} cells, which are all higher expressed in CLL LNs in comparison to rLNs. In contrast to that, we observed a lower expression of the activation marker CD38 on CD8 T_{EM} cells in CLL vs rLN.

New Figure 3E:

[editorial note: redacted by request of authors]

2. To me, the reason why anti-PD1 does not work in CLL might arise from studying Richter's transformation, where anti-PD1 is efficacious in some patients. Do the authors have access to RT patients LN or some other tissue, or are there perhaps already published data?

Response: We agree with you, that comparing the T-cell landscape in CLL and Richter's transformation is highly interesting. Unfortunately, we do not have access to LN samples from patients with RT. But there is published scRNA-seq data of 17 bone marrow samples longitudinally collected from 6 patients with RT (Parry et al., Cancer Cell 2023). The main finding of this study is the detection of an intermediate exhausted CD8 effector/effector memory T-cell cluster marked by high expression of ZNF683 that is associated with response to anti-PD1 therapy. We have analyzed the expression of ZNF683 in our scRNA-seq data and identified two very small clusters of cells expressing this gene (see Figure below). One cluster falls into the CD8 T_{EM} cells, and the other into the CD4 T_{FH} cells. The CD8 T_{PEX} cluster identified in our data does not show any ZNF683 expression. This suggests that the progenitor or intermediate states of T-cell exhaustion are different in CLL and RT. One major limitation of this comparison is that the published RT data is from bone marrow and our scRNA-seq data set is from LN samples, but it still allows to speculate that the phenotype and/or frequency of early exhausted T-cell subsets that are known to be required for response to ICI are different in CLL compared to RT. We have now included a paragraph on these findings in the Discussion of the manuscript.

3. Clinical association and LN immune cells. With such a small sample size ($n=20$), one perhaps would not expect to find meaningful association after the adjusted P values. Although I applaud the authors for the correct way of doing the adjusting, were there any findings that were significant before the comparisons? Especially interesting would be to see some effect size, e.g., log₂ fold-change between the clinical categories.

Response: We would like to thank you for this suggestion. We are now providing correlations of all cell clusters with clinical data before and after adjustment for multiple testing (see new Suppl. Table 4). This shows that several T-cell clusters are associated with tumor load, Rai or Binet stage before multiple testing. In addition, we observed a significant correlation between CD4 T_{CM3} cells and time to treatment, as well as CD4 T_{FH} cells and survival. We provide further a new Kaplan Meier analysis showing a significant survival benefit for patients with a higher abundance of CD8 T_{EM} GZMK cells (see new Suppl. Figure 4G).

New Suppl. Figure 4G:

[editorial note: redacted by request of authors]

4. TP53/del 17p is clinically one of the most interesting populations. I see the authors have only three such samples, BC1, BC12 (and BC15?), but for many samples the analysis is not available. Could the authors retrieve TP53/del17p status for all the samples? While clinically a physician might not study the TP53 study if they are not planning a treatment, at least the treated patients should have this information available. There are many reasons to believe that patients with these abnormalities might have a distinct immune system that is more malleable with immunotherapies.

Response: Thanks for this helpful comment. We have now included the TP53/del17p status of all patients (updated Suppl. Table 1) and specifically compared samples with and without these aberrations in terms of T-cell landscape. Altogether, 3 patients with del17p were included in our cohort. And even though the LN samples of these cases were not considerably different compared to the non-affected samples, this was different in the PB samples. As shown in Suppl. Fig. 4A, all 3 samples with del17p (BC1PB, BC12PB and BC15PB) clustered together and were different from the other 7 PB samples. The main distinction was an enrichment of CD4 T_{EM} CD39 cells (orange cluster) that was highly abundant only in these three samples. In addition, the del17p samples contained significantly more CD4 T_{EM} GZMK, T_{PR} and DN T_{EM} HELIOS cells compared to TP53 wild-type cases. We have added a new Suppl. Figure 4G and the description of this interesting finding to the manuscript, highlighting also current literature on this topic.

New Suppl. Fig. 4H:

[editorial note: redacted by request of authors]

5. Comparing T cells phenotypes from LN to PB with TCRseq. Majority of CLL studies, have focused on circulating CLL and non-CLL immune cells. As the authors have from some individuals both PB and LN, it would be crucial to deliver what are the disease biomarkers that are only seen in LN, what are seen in PB, and what are seen in both. To me, this should be done with scRNA+TCRab-seq. Could the authors do additional scRNA+TCRab-seq to follow T cells clones from lymph node in peripheral blood? Of course following these T cells in bone marrow would be optimal but the process might be too invasive. Especially the BC1 with TP53mut and hyperexpanded clone would be informative.

Response: We agree that comparing single-cell-resolved transcriptomes and clonality of T cells from CLL blood and lymph node samples is of high interest. That is why we performed additional experiments adding scRNA+TCR-seq of 5 CLL PB samples, of which we already had corresponding data of LNs. This data is now presented in the new Figures 4A-C and 5A-D and Suppl. Figure 7A-D, and described on pages 8-10 of our manuscript (see highlighted passages). The comparative analysis of this data allowed us add the following highly relevant conclusions to our manuscript:

- (1) In comparison to paired CLL PB samples, CLL LNs are enriched in regulatory T cells and CD8 exhausted T cells, whereas CD8 T_{EM}, NK-like and MAIT cells are higher abundant in PB versus LN samples.
- (2) T cell expansion is more prominent in PB compared to LN, but comparable T-cell subsets are affected in both compartments.
- (3) Shared TCR clones exist in PB and LN samples of the same patient, but the size of shared clones is not the same both compartments. Treatment of patients reduces the frequency of shared clones.

New Figure 4A-C:

[editorial note: redacted by request of authors]

New Figure 5A-D:

[editorial note: redacted by request of authors]

New Suppl. Figure 7A-D:

[editorial note: redacted by request of authors]

6. T cell antigens. Do the authors have any hints what are their most interesting T cells targeting? Tumor associated antigens, neoantigens (somatic mutations / CNVs); what are known, what are the authors thoughts? Please discuss. Could the authors do some unsupervised clustering of the TCRs within / across individuals to have an intuition how many antigens the TCR might be targeting? Some algorithms could be e.g., GLIPH2, TCRdist.

Response: We recognize the significance of the questions raised. To address this comment, we utilized the GLIPH2 algorithm alongside predicTCR to explore the reactivity of T cells in CLL. The results of these analyses are now presented as new Figure 5E-G and Suppl. Figures 7E-H and 8, and described on pages 10-11 of our manuscript.

The recently published predicTCR algorithm (Tan et al., Nat. Biotech. 2024) allowed us to identify numerous T-cell clusters with a highly predicted tumor reactivity. The predicted CLL-reactive T cells were more frequent in LN compared to PB samples, and enriched in the clusters of CD8 exhausted T cells, proliferating T cells, follicular helper and regulatory T cells. Integration of TCR-seq data showed that CLL-reactive clones were small and resided either only in the LNs or were detectable in LN and PB samples. T-cell clones that were detected only in PB were mostly non-reactive. Altogether, this suggests that CLL-reactivity of T cells is predominantly happening in LNs with a low circulation rate of these cells to PB. Notably, within some clusters predicted as non-tumor-reactive, TCRs display characteristics consistent with CMV reactivity, supported by both existing literature and HLA matching with entries in the VDJdb and ImmuneDETECT databases. We have now included a discussion section exploring potential CLL T-cell antigens.

New Figure 5E-G:

[editorial note: redacted by request of authors]

New Suppl. Figure 7E-H:

[editorial note: redacted by request of authors]

New Suppl. Figure 8: please refer to the submitted manuscript

7. CLL LN vs healthy LN scRNA-seq. It is known that CD4+ T cells do not expand as much as CD8+ T cells, so I would say that 4-26 % of clonal expansion of CD4+ T cells are major, unlike the authors. Could the authors compare their CLL LN scRNA+TCRab-seq data to healthy LN scRNA-seq data, like they have in the interaction analyses? This could put the phenotypes into perspective, and hopefully the clonal

expansion rates. Ideally these samples should be somewhat age-matched, as that is the major determinant.

Response: To follow your interesting suggestion, we have now analyzed scRNA-seq data of 3 LN samples from healthy donors published by Roider et al. (Nat. Cell. Biol. 2024). Based on the transcriptome data, we have defined and quantified the main T-cell clusters in these samples, and retrieved T-cell clonality from the TCR-seq data. Similar, as in the CLL LNs, there is a higher degree of clonal expansion of CD8 T cells compared to CD4 T cells, which showed a clonal expansion of 4, 5 and 12% in comparison to 19, 37 and 43% of CD8 T cells (see Figure below). In general, the extent of clonal expansion, both in the CD4 (ranging from 2-12%) and the CD8 (ranging from 12-43%) T-cell compartment was comparable in CLL and rLNs, with the exception of BC1 showing a hyperexpanded CD8 T-cell clone, leading to a total expanded CD8 T-cell population of 67% (see Figure 5A). An interesting difference between CLL and rLNs is the phenotype of the expanded cells. Whereas in rLNs mainly CD8 T_{EM} cells are clonally expanded, we identified also CD8 T_{PEX} and NK-like cells as clonally expanded cells in CLL LNs. This can be likely explained by the acute versus chronic exposure of T cells to antigen in rLN versus CLL LNs, respectively.

To address your comment and better describe our observations, we have now changed the text of our manuscript stating now:

Updated Results on page 10: Thereby, we identified a major clonal expansion of CD8+ T cells (ranging from 12% to 76%) and a lower expansion rate of CD4+ T cells (2% to 38%; Figure 5A). The expansion of T cells was significantly higher in PB compared to LN samples (Suppl. Figure 7A), and we did not observe a correlation between the expansion rate of CD4+ and CD8+ T cells.

8. LGALS9 in stead of TIM3. Why would the authors target LGALS9 instead of TIM3, which is an immune checkpoint and is/was of great interest for hematologists in MDS/AML with sabatolimab? To me, the data analysis on solid cancers is obscure and the OS data with 50 CLL samples is not convincing that LGALS9 is a better target.

Response: The results of our interactome analysis identifies LGALS9 as interaction partner for several target molecules (P4HB, HAVCR2, PTPRC, CD47, CD45, CD44; see Figure 6C and F). That is why we decided to block Galectin-9 instead of TIM3, which is only one of the potential interaction partners. In the TCL1 mouse model of CLL, we detected an upregulation of Galectin-9 and TIM3 with leukemia development. TIM3 expression was induced in several T-cell subsets, including CD8 T_{EF} and CD4 T_{REG}, up to 30% of the cell subset. In contrast, Galectin-9 was detectable in more than 90% of all CLL cells. Altogether, these findings made Galectin-9 a more abundant and suitable target for immunotherapy. In the meantime, there is also evidence that suggests Galectin-9 as a novel immunotherapeutic target in follicular lymphoma (Blood Cancer J. 2024 May 2;14(1):7) and many other cancer entities (reviewed in Front Cell Dev Biol. 2024 Jan 9;11:1332205). Besides our own results, there is published data linking Galectin-9 expression with Treg expansion (FASEB J. 2021 Jul;35(7):e21556) and outcome of patients in CLL (Cancers (Basel). 2023 Nov 11;15(22):5370; Immun Inflamm Dis. 2023 May;11(5):e853).

9. What are the main points learned from the IHC? To me, it seems that these data are underused.

Response: The main advantage of multiplexed spatial analyses of tissues is to not only quantify immune cells subsets but also explore their context and neighborhood in the tissue. We assessed the spatial cell-cell interactions by calculating the enrichment of pairwise interacting cell types for all cores using the Giotto platform. By this, we identified an enrichment of physical interactions between CD8 PD1+ T cells as well as CD4 PD1+ T cells with CD4 T_{REG}. This data suggests that antigen-activated T cells (PD1 is induced upon activation of CD4 and CD8 T cells) are in direct contact with T_{REG} in CLL lymph nodes, which like suppress and limit their activity contributing to immune escape.

Updated manuscript text: We next quantitatively assessed the spatial cell-cell interactions by calculating the enrichment of pairwise interacting cell types for all cores using Giotto¹⁸. This confirmed an enrichment of physical interactions between CD4 T_{REG} and CD8 PD1+ T cells as well as CD4 PD1+ or PD1- T cells (Figure 2C-D, Suppl. Figure 3D) suggesting that T_{REG} limit the activity of CD4 and CD8 T cells in the CLL LNs.

Corresponding Figure 2:

C) Representative CLL LN-derived tissue core (left image) and field of view (middle image) displaying FOXP3 (yellow), CD8A (pink), PD-1 (light blue) and DAPI (blue) staining. The right image displays cells identified as TREG and CD8 PD-1+ T cells in yellow and purple dots, respectively. D) Boxplots (left) depicting the range of enrichment of pairwise interacting cell types for all cores (n = 42). Boxplots (right) showing the absolute number of interactions in log10 scale between pairwise interacting cell types across all cores. Colored by type of interaction, between same cell type (homotypic, blue) or different cell types (heterotypic, red). Enriched heterotypic interactions with TREG are marked in bold.

Corresponding Supplementary Figure 3D:

D) Heatmap showing the significance score of pairwise interacting cell types (columns) for each core (rows) (n = 42) as generated by Giotto. Cell type pairs and cores were both grouped based on hierarchical clustering.

Minor:

1. Discussion / intro related to CLL therapies. The most common CLL therapies; Obi-Ven, BTK, FCR; have distinct effects on the immune system - could the authors discuss how their findings are attributed to these?

Response: Thank you for raising this interesting question. We agree that it is important to consider the impact of current CLL therapies on the immune microenvironment and specifically on T cells, since the majority of the drugs that are in use, are impacting not only on B cells but also on T cell function. This question is of most interest when it comes to developing combination therapies or sequential treatment including immunotherapeutic approaches. Due to limited word counts, we were able to only add minor statements and comments in our revised version of the manuscript to address this important question.

Updated Introduction on page 4: In addition, current treatment regimens for CLL were shown to impact T-cell fitness⁷.

2. Discussion / intro related to ICI. I think the field has moved on from ICIs towards bispecifics and CAR T:s, with and without combinations (e.g., BTKs). I think the intro / discussion should reflect more of that, although I do understand that readers of Nat Comm might be more accustomed to ICIs.

Response: We agree that our findings are of specific interest in the context of novel treatment approaches including bispecifics and CAR-T cells. We have briefly included these regimens now also to the discussion of the manuscript.

Updated Discussion on page 15: T cell exhaustion is considered as one of the main reasons for failure of immunotherapies, including immune checkpoint inhibitors, CAR-T cells or bispecific T-cell engagers.

3. LNs from same individuals. Could you comment on the LNs from the same donor? These are best biological replicates, as the host genetics and disease covariates are identical.

Response: We appreciate the referee's comment about the value of paired samples, particularly those collected at different stages of disease progression. However, with only two such samples available, drawing firm conclusions would be difficult. Therefore, we decided not to include the analysis presented below.

For case BC5LN, samples were taken 7 years apart, with treatment initiated 1.5 years before the first sample. Here, we observed an increase in naïve CD4 and CD8 T cells at time point 2 (BC15LN2), while CD8 T_{EM} GZMK and T_{EM} ICOS cells decreased (see Figure below).

For case BC15LN, samples were taken 1.8 years apart, and the patient had not received any treatment before the samples were taken. In this case, there was an increase in CD8 T_{EX}, CD4 T_{EM} GZMK, and T_{REG} PD1 T cells (see Figure below).

4. What is the main new information generated with the pseudotime analyses? What are the information what could've not been received otherwise, are there some therapeutic applications, is the data valuable with some in vitro / in vivo assays?

Response: The pseudotime analysis suggests a trajectory of T-cell subsets which allows us to conclude that CD8 T_{EM} GZMK cells are precursor cells of terminally exhausted T cells. As this T-cell population is linked to better outcome, we assume that these cells are CLL-reactive and control disease, but the continuous exposure to antigen drives them towards terminal exhaustion. A similar exhaustion pathway is also suggested for CD4 T cells. As T-cell exhaustion is hard to model in vitro, in vivo cell tracing assays would be the best option for validation. But as these experiments are technically challenging and time-consuming, they are beyond the scope of this manuscripts.

5. Cell types enriched in rLN. It would be helpful to at least in the supplements see what are the cell types that are enriched in rLN compared to CLL LN. For example, are TFH enriched to rLN?

Response: We appreciate this comment and have now added a new Suppl. Figure 4C showing all cell subsets that are enriched in rLN vs CLL LNs.

Supplementary Figure 4C:

C) Boxplot showing cell subset abundances out of total T cells in CLL LNs (n = 20) and rLN (n = 13).

6. TIM3 in CyTOF. I see that in the methods that the signal for TIM3 was too low to be acceptable for downstream analysis. Does that mean that it is not expressed enough or that the antibody does not work?

Response: Unfortunately, the TIM3 antibody works only with fresh samples but not with viably frozen cells. Since we performed CyTOF with cryopreserved patient samples, the lack of TIM3 detection is technical.

7. Lines 68 - 69: I find these lines confusing. I think the authors mean SLL and not CLL. Please revise.

Response: Thanks for this comment. We have modified the text to avoid confusion.

Updated Introduction on page 4: Very low response rates to anti-PD1 antibodies were observed in patients with chronic lymphocytic leukemia (CLL)^{1,2}, a malignancy of mature B cells that accumulate in blood, LNs, and bone marrow.

8. Could NCAM be referred as CD56? I think people are more accustomed to that naming.

Response: We have changed NCAM to CD56 throughout the text and figures.

9. Line 430 Galectin-9

Response: Thanks for pointing this out. This has been changed in the revised version.

10. Figure 1A and C require greater resolution

Response: Due to size limitations, we could not upload high-quality figures to the platform of the journal. The problem with the resolution of the figures is solved in the final version of the manuscript.

11. Figure 3B would benefit from adding hovered broad labels for the islands, such as CD8+, CD4+ etc, know skipping between Fig 1 and 3 is bit cumbersome.

Response: Thanks for this suggestion. We have added now an extra UMAP to Figure 3B (see below) that shows the major T-cell clusters with labels for easier comprehension.

New Figure 3B:

[editorial note: redacted by request of authors]

Reviewer #3 (Remarks to the Author):

In this paper by Llào Cid and colleagues, the Authors investigated, by complementary approaches, the T cells landscape in the CLL microenvironment. The characterization of these subsets of cells allowed to identify a disease-specific accumulation of regulatory and exhausted populations. Moreover, by looking at the interactions between CLL and T cells, the Authors reported the identification of Galactin-9/TIM3

axis as relevant in driving and shaping CLL immune-suppression and immune-escape.

The paper is well written and organized. The reported findings are novel and provide significant insight in the understanding of CLL pathogenesis and responses to therapies, opening for novel targeting opportunities.

However, there are several points that need to be addressed by the Authors.

Response: Thank you very much for the thorough review of our manuscript and this positive comment. We are happy that you appreciate the potential of our novel data in terms of disease biology and translational opportunities.

Major points.

1. How do the Authors explain the low levels of TIM3 transcripts and its expression as a receptor expressed in exhausted T-cells? Are there any post-transcriptional mechanisms involved in their expression?

Response: Based on our scRNA-seq data, the average transcript level of TIM3 in exhausted CD8 T cells is high, but it is expressed only in a subset of cells (~50%), likely marking the dysfunctional cells. Unfortunately, the TIM3 antibody works only with fresh samples but not with viably frozen cells. Since we performed CyTOF with cryopreserved patient samples, the lack of TIM3 protein detection in our analysis is technical.

2. Assuming that there are differences in the ratio between regulatory and inflammatory blood cells between males and female specimens - as shown in the literature – is there a potential bias in using only female mice? This point has to be discussed.

Response: Thank you for bringing up this interesting point. The reason for using only female mice in our in vivo studies, is a more homogeneous disease development upon TCL1 cell transplantation in those compared to male mice. This allows for detecting significant differences in CLL development in cohorts of 10 mice per treatment arm, which would not be possible in male mice. To get insights whether the sex impacts on the T-cell compartment in CLL, we performed correlation studies of all T-cell clusters with sex and several other clinical features. According to the suggestion of referee #2, we have included now correlation analyses with and without multiple testing (Suppl. Table 4). We did not observe any significant correlations of T-cell clusters with sex according to the adjusted p-value after multiple testing, and only a significant correlation of naïve CD8 T cells with sex without multiple testing. This analysis suggests that there is no major sex bias in terms of the T-cell landscape and its therapeutic targeting in CLL. We have included the following brief statements on this in the Results and Methods sections of the revised manuscript.

Updated Results on page 7: We did not observe significant associations between the frequencies of T-cell clusters with sex, clinical stage, or outcome of the patients according to the adjusted p-values after multiple comparisons (Suppl. Table 4).

Updated Methods on page 18: The use of female mice was justified due to a more homogeneous disease development in those compared to male mice, and due to no major differences observed in the T-cell landscape of male and female CLL patients.

3. In Figure 1B, in the UMAP plot, in the bottom left corner (-5 and -5 as coordinates) there is a little cluster of cells with the same color of cluster 1. Is it just noise or has the first cluster a subpopulation which is separated from the rest? If so, please comment and explain why they are distinct.

Response: Thank you for pointing this out. The small separate cluster is highly similar to the rest of cluster 1, clearly fitting to the definition of naïve CD4 T cells, besides high TIGIT expression. As TIGIT is induced in activated T cells and high expression is not expected in naïve CD4 T cells, we assume that this sub-cluster is likely technical noise.

4. In Figure 1D, it seems that there is a pattern of exhaustion in T-reg cells instead of cytotoxic/effector T-cells. Referring to the statement reported at line 145-150, there seems to be a strong correlation between CD4+ T-reg and PD1+ CD8+ T-cells, which is in line with a role of CD4+ T-regs in promoting the accumulation of dysfunctional CD8+ T-cells. Do the Authors suspect the existence of a crosstalk between regulatory and cytotoxic cells as a primary cause of the exhausted phenotype? Indeed, the persistent antigenic stimulation, as well as a general transcriptional reprogramming, is regarded to be one of the main causes for T-cell exhaustion.

Response: We agree that indeed a crosstalk between regulatory and cytotoxic T cells is very likely and suggested to contribute to T-cell exhaustion in cancer. In Figure 6A, we show the results of an interactome analysis, showing that T_{REG} as sending cells are predicted to strongly interact with CD8 T cells, from naïve to T_{EM}, T_{PEX} and T_{EX} cells. And when focusing on interactions that are differential between CLL LNs and rLNs, the highest number of predicted interactions of T_{REG} is with CD8 T_{EM} (Figure 6D). As the interactome analysis predicted the highest number of interactions between CLL cells and CD8 T cells, we focused in our study on those. But we agree, that it is of high interest to further analyze the predicted interactions between T_{REG} and CD8 T cells as follow-up work, as these likely contribute to immune escape.

5. Referring to Figure 2B, although a really low p-value is calculated, I would not state that there is a strong correlation since the R2 is 0.37. The correlation is there, but not that strong. I would rephrase accordingly.

Response: Thanks for pointing this out. We have changed the text deleting the statement that the observed correlation is strong.

Updated Results on page 6: Quantification of the frequencies of the major immune cell subsets in the tissues revealed a positive correlation for CD8+ PD1- and CD8 PD1+ T cells, for myeloid cells and CD4 T_{REG}, and for CD4 T_{REG} and CD8+ T cells (Figure 2B, Suppl. Figure 3C) which is in line with a role of CD4 T_{REG} in promoting the accumulation of dysfunctional CD8+ T cells¹⁷.

6. The Authors focused on CD4 PD1+-CD4 Tregs and CD8 PD1+-CD4 Tregs interaction, however, as shown in Figure 2D, there are other cells involved in heterotypic cell-interaction with greater/similar enrichment (e.g. CD4 PD1--CD4 Tregs). Why did the Authors focus only on the first one? Please provide an explanation.

Response: This is a valid statement. We have changed the manuscript now including also the interactions between CD4 T_{REG} with CD4 PD1⁻ cells. The focus remains on T_{REG} as they are enriched in CLL LNs and predicted to strongly interact with other T cell subsets. In addition, T_{REG} are known mediators of immune escape, and there is published data showing that depletion of T_{REG} improves immune control and slows down disease development in CLL mouse models.

Updated Results on page 6: This confirmed an enrichment of physical interactions between CD4 T_{REG} and CD8 PD1⁺ T cells as well as CD4 PD1⁺ or PD1⁻ T cells (Figure 2C-D, Suppl. Figure 3D) suggesting that T_{REG} limit the activity of CD4 and CD8 T cells in the CLL LNs.

7. In Figure 4D-E: are the statistical differences observed also statistically assessed? In other words, did the Authors also conduct a statistical analysis or is the significance already computed by the scoring methods?

Response: Thank you for raising this question. Kruskal-Wallis rank sum test, together with pairwise comparisons using Wilcoxon rank sum test with corrections for multiple testing (BH) were used. Stars in Figures 4D and E indicate that the CD8 T_{EX} and CD8 T_{PEX} subsets have statistically significantly higher signature scores compared to all other subsets. The details are now described in the figure legend:

Updated legend to Figure 4D-E: Violin plot of average expression levels in LN T-cell subsets of the exhaustion gene signature adapted from Zheng L, et al Science, 2021 (D), and the precursor exhaustion gene signature from Guo X, et al Nat Med, 2018 (E). Stars indicate that the CD8 T_{EX} (D) and CD8 T_{PEX} (E) subsets have statistically significantly higher signature scores compared to all other subsets.

*Statistical significance was tested by unpaired t-test, (C) and Wilcoxon rank sum test (D, E), *p<0.05, **<0.01, *** < 0.001.*

8. Referring to Figure 5A, as only 5 patients were evaluated for the remaining part of the study, did the Authors use any tool to assess the statistical power of this setting? Were they the only patients at disposal? Was the choice of patients randomized? Did the Authors assess the presence of any confounding factors for the study? For instance, are there any genetic variants likely implicated in the crosstalk with the TME?

Response: As LN samples from CLL patients are hardly available, the selection of 5 cases for scRNA- and TCR-seq was based on availability of viably frozen single-cell suspensions containing sufficient numbers of T cells. Unfortunately, we could not randomize or assess the statistical power of our data. To better assess the presence of confounding factors, we are providing now an updated Suppl. Table 1 including all available clinical information of the patients involved in our study, and are using this data to analyze correlations of all T-cell clusters with clinical data before and after adjustment for multiple testing (see new Suppl. Table 4). This shows that several T-cell clusters are associated with tumor load, Rai or Binet stage and outcome of patients before multiple testing. We further specifically looked at a potential impact of TP53/del17p status on the T-cell landscape. Altogether, 3 patients with del17p were included in our cohort. And even though the LN samples of these cases were not considerably different compared to the non-affected samples in the CyTOF analysis, the PB samples of these 3 cases appeared as distinct. As shown in Suppl. Fig. 4A, these 3 samples with del17p (BC1PB, BC12PB and BC15PB) clustered

together and were different from the other 7 PB samples. The main distinction was an enrichment of CD4 T_{EM} CD39 cells (orange cluster) that was highly abundant only in these three samples. In addition, the del17p samples contained significantly more CD4 T_{EM} GZMK, T_{PR} and DN T_{EM} HELIOS cells compared to TP53 wild-type cases (Suppl. Fig. 4G). For one of the three patients with del17p, we also obtained scRNA-seq data of the LN sample (BC1). Interestingly, this sample showed the highest degree of clonal expansion of CD8 T cells, with a large hyperexpanded clone. We have added a description of these interesting findings to the manuscript, highlighting also current literature on this topic (please see below). But due to the small number of analyzed cases, there are no major statements allowed concerning the impact of genetic variants on the TME.

Updated Results on page 7: We did not observe significant associations between the frequencies of T-cell clusters with sex, clinical stage, or outcome of the patients according to the adjusted p-values after multiple comparisons (Suppl. Table 4). However, significant associations between several cell clusters and tumor load or clinical stage were detected before multiple testing (Suppl. Table 4; p-value <0.05). In addition, we observed a significant correlation between CD4 T_{CM3} cells and time to treatment, as well as CD4 T_{FH} cells and survival. A Kaplan-Meier- survival analysis showed a significant benefit for patients with a higher abundance of CD8 T_{EM} GZMK cells (Suppl. Figure 4G), suggesting their involvement in CLL control. As 3 of the patients harbored a deletion in 17p, including the TP53 locus, we specifically compared those cases with TP53-proficient samples. Even though the LN samples of these cases were not distinct, all 3 PB samples with del17p (BC1PB, BC12PB, and BC15PB) clustered together due to a higher abundance of CD4 T_{EM} CD39 cells (Suppl. Figure 4A, orange cluster). In addition, the del17p samples contained significantly more CD4 T_{EM} GZMK, T_{PR}, and DN T_{EM} HELIOS cells compared to TP53 wild-type cases (Suppl. Figure 4H). This is in line with emerging knowledge that TP53 modulates tumor immunity²⁰.

9. Referring to Figure 5B: what is the reason behind the NA values? Was the identification of the markers of clonal expansion not definable?

Response: Yes, the NA values were assigned to cells for which no TCR information was available. We have now added this information to the figure legend.

Updated legend to Figure 5B: UMAP plot colored according to the T-cell clone size based on the TCR-seq data. NA: no TCR information available.

10. Referring to Figure 7B: how do the Authors explain the fact that CD4 T_N cells present similar level of expression markers to CD8 T cells?

Response: We agree that CD4 T_N and CD8 T_N are similar in respect to the 3 selected naïve markers that we show in the dot plot in Figure 7B (Ccr7, Lef1, Il7r). We are presenting the expression of these markers as they are frequently used to identify naïve T cells. But as one can appreciate in the UMAP shown in Figure 1A, CD4 T_N and CD8 T_N are clearly distinct transcriptome-wide, as they cluster separately as cluster 1 and 2+9+11, respectively.

11. In Figure 8A data are reported as TPM, but it would be better to plot them as TMM values. Indeed,

TPMs are best suited for intra-sample comparisons. Alternatively, could the Authors provide a statistical test to assess the differences between healthy and patients?

Response: We are sorry for not describing this clearly. We have used the GEPIA2 platform with default settings to assess differential expression of *LGALS9* in TCGA data (see: <http://gepia2.cancer-pku.cn/#general>). All TCGA data analyzed by GEPIA2 are provided in Suppl. Figure 9A. Figure 8A shows only those cancer types that showed a differential expression of *LGALS9* between tumor and normal tissue. We have included this information now in the figure legends as follows:

Updated legend to Figure 8A: Differential expression of LGALS9 in tumor versus healthy tissue in CESC (Cervical squamous cell carcinoma and endocervical adenocarcinoma), COAD (Colon adenocarcinoma), DLCL (Diffuse large B cell lymphoma), ESCA (Esophageal carcinoma), GBM (Glioblastoma multiforme), KIRC (Kidney renal clear cell carcinoma), KIRP (Kidney renal papillary cell carcinoma), LAML (Acute Myeloid Leukemia), LGG (Brain Lower Grade Glioma), LIHC (Liver hepatocellular carcinoma), OV (Ovarian serous cystadenocarcinoma), PAAD (Pancreatic adenocarcinoma), READ (Rectum adenocarcinoma), SKCM (Skin Cutaneous Melanoma), STAD (Stomach adenocarcinoma), TGCT (Testicular Germ Cell Tumors), and UCEC (Uterine Corpus Endometrial Carcinoma) analyzed by the standard processing pipeline GEPIA2 with default cut-off settings⁴⁸.

*Updated legend to Suppl. Figure 9A: Comparative analysis of LGALS9 transcript levels in tumor versus healthy tissue in ACC (Adrenocortical Carcinoma), BLCA (Bladder Urothelial Carcinoma), BRCA (Breast invasive carcinoma) CESC (Cervical squamous cell carcinoma and endocervical adenocarcinoma), CHOL (Cholangiocarcinoma), COAD (Colon adenocarcinoma), DLBC (Lymphoid Neoplasm Diffuse Large B-cell Lymphoma), ESCA (Esophageal carcinoma), GBM (Glioblastoma multiforme), HNSC (Head and Neck squamous cell carcinoma), KICH (Kidney Chromophobe), KIRC (Kidney renal clear cell carcinoma), KIRP (Kidney renal papillary cell carcinoma), LAML (Acute Myeloid Leukemia), LGG (Brain Lower Grade Glioma), LIHC (Liver hepatocellular carcinoma), LUAD (Lung adenocarcinoma), LUSC (Lung squamous cell carcinoma), MESO (Mesothelioma), OV (Ovarian serous cystadenocarcinoma), PAAD (Pancreatic adenocarcinoma), PCPG (Pheochromocytoma and Paraganglioma), PRAD (Prostate adenocarcinoma), READ (Rectum adenocarcinoma), SARC (Sarcoma), SKCM (Skin Cutaneous Melanoma), STAD (Stomach adenocarcinoma), TGCT (Testicular Germ Cell Tumors), THCA (Thyroid carcinoma), THYM (Thymoma), UCEC (Uterine Corpus Endometrial Carcinoma), UCS (Uterine Carcinosarcoma) and UVM (Uveal Melanoma) using the standard processing pipeline GEPIA2 with default cut-off settings⁴⁸. Tumors with differential expression of *LGALS9* are indicated in red.*

12. Data included in Figure 6B are redundant with data plotted in Figure 6A: please remove them.

Response: We agree with the referee that the information we have plotted in Figure 6B was already included in Figure 6A indicated as bars at the edges of the plot that display the number of interactions for all cell clusters. As we think that this information can be easier captured from Figure 6B, we suggest to keep this figure, but have rather removed the bars in Figure 6A. Thereby, we avoid redundancy in the display of our results and hope that this solution is acceptable for you.

13. In the last section of the results, the Authors look at *LGALS9* expression in other tumors than CLL.

This part is somehow out of the topic of the paper, that is centered on CLL, and it would be better to keep it as a point of discussion, instead of results.

Response: It is true that our manuscript focusses on CLL. But LGALS9 seems to have a broader relevance as immunomodulatory molecule and therefore, potential treatment target in other cancer entities, we think that the data shown in Figure 8 are of relevance. Considering that Nature Communications targets a broader audience than scientists interested in blood cancers, we think that this part of the manuscript is interesting for the readers and therefore would like to keep it in the paper.

14. Given the interesting finding of galactin-9, it would be great to include this finding in the title of the paper.

Response: We would like to thank the referee for this suggestion. We have added Galectin-9 as novel immunotherapy target in the title of the manuscript.

Updated Title: Integrative multi-omics defines the T-cell landscape in CLL and identifies an enrichment of regulatory and exhausted T-cell types and Galectin-9 as novel immunotherapy target

Minor points, mainly related to graphical issues.

1. Figures 1A, C and 7E are blurred.

Response: Due to size limitations, we could not upload high-quality figures to the platform of the journal. The problem with the resolution of the figures is solved in the final version of the manuscript.

2. Figure 4G has “z-scores” cut in half.

Response: Thanks for pointing this out. This has been changed in the revised version.

3. Figure 2C has no piece of information about the blue staining (likely a DAPI for the nucleus)

Response: Thanks for pointing this out. We have added this information to the figure.

POINT-BY-POINT RESPONSE LETTER TO REVIEWER COMMENTS

Reviewer #2 (Remarks to the Author):

I thank the authors for the responses to my comments. The following emerging issues need to be addressed:

1. Please tone down the claims of identifying CLL-reactive T cells based on analyzing gene expression profiles. The claims in their current forms are truly misleading, and the authors of predicTCR also recommend using their tool as a mere prioritization strategy for further antigen-reactivity screening. The analysis presented here is highly circular – predicTCR looks for gene expression of exhaustion related genes, so it is not surprising that the authors find the most “tumor reactivity” in cells expressing exhaustion related genes and that they are predominantly found in LN compared to PB. One of the main differences is also limited that predicTCR is trained on TILs, and I would argue that T cells in CLL LN are not the same as TILs, as the authors also state in their response to me.

The analysis in the current form is not in line with what I asked, where I inquired about similarity within TCRs. The lines 334-351; are more in line with what I ask, but would not highlight these in the abstract and would remove wild claims related to predicTCR. Especially lines 335-338 and 346-347 need to be removed, as I cannot understand how the authors claim validation for such a method without doing antigen-reactivity.

To be clear, I would remove the analyses performed with predicTCR as circular. Or then validate some of their hits with true antigen screening (e.g., by cloning TCRs and testing if they indeed react against the autologous CLL cells), but as hinted by the authors in the Discussion that is not in the immediate plans.

2. I believe data should be available in a non-raw format (e.g., gene-cell matrices) and additionally code should be also available.

Response: We thank the reviewer for their continued detailed assessment and constructive feedback. We have carefully revised the manuscript in response to all points raised.

1. Regarding the use of predicTCR and interpretation of CLL-reactive T cells:

We fully acknowledge the limitations of using gene expression profiles alone to infer antigen specificity. In the revised manuscript, we have significantly toned down the claims related to identifying CLL-reactive T cells. We now clearly present predicTCR as a *prioritization* tool rather than a validation of tumor reactivity, and have aligned our interpretation with the recommendations made by the original authors of predicTCR.

Specifically:

- We have removed the statements in lines 335–338 and 346–347 that the reviewer correctly identified as overreaching.
- We have reworded sections of the Abstract and Discussion to eliminate any implication that we validated CLL-specific TCRs or that predicTCR directly confirms tumor reactivity.
- We retained the gene expression-based prioritization analyses as an exploratory tool, clarifying that these findings are *hypothesis-generating* rather than conclusive.

- We also emphasize that predicTCR is not solely based on transcriptome data but integrates both transcriptomic features and VDJ-sequencing data (i.e., CDR3 sequences and TCR gene usage), providing a more comprehensive profile for TCR prioritization.
- We agree with the reviewer that T cells in CLL LNs are not equivalent to classical TILs and now emphasize this distinction more clearly in the manuscript.

Given the significant technical limitations we encountered (detailed in our response to the Editor), we were not able to perform antigen-specific validation of individual TCRs at this stage. We now explicitly state this limitation and have refrained from making any claims of functional validation.

2. Regarding data and code availability:

Human single-cell RNA- and TCR-sequencing data including matrix files are deposited to European Genome-phenome Archive (EGA) under the accession ID EGAS00001006864 and are available on request. Mouse single-cell RNA- and TCR-sequencing data are available on Gene Expression Omnibus under the accession ID GSE221395. All code related with CyTOF and scRNA-seq analyses is available on GitHub (https://github.com/mzapatka/Tcell_CLL). This ensures full transparency and reproducibility of our analyses.

We appreciate the reviewer's feedback, which has led to a more cautious and accurate presentation of our results.

Reviewer #3 (Remarks to the Author):

The Authors have addressed all the issues raised and modified the paper accordingly.

No more points from my side.

Response: We thank the reviewer for their positive evaluation and appreciation of our revisions.

Point-by-point response to REVIEWERS' COMMENTS

Reviewer #2 (Remarks to the Author):

I thank the authors for their revised manuscript. They have addressed all my concerns.

Response: thanks very much for your help in reviewing and improving our manuscript.